# Real-time dispersal of malaria vectors in rural Africa monitored with lidar

**Samuel Jansson**[1,2]\*, **Elin Malmqvist**[1,2], **Yeromin Mlacha**[3,4,5], **Rickard Ignell**[6], **Fredros Okumu**[3,7,8], **Gerry Killeen**[3,9], **Carsten Kirkeby**[10,11], **Mikkel Brydegaard**[1,2,11,12]

**1** Lund Laser Centre, Department of Physics, Lund University, Lund, Sweden, **2** Center for Animal Movement Research, Department of Biology, Lund University, Lund, Sweden, **3** Environmental Health and Ecological Sciences Department, Ifakara Health Institute, Ifakara, Tanzania, **4** Swiss Tropical and Public Health Institute, Basel, Switzerland, **5** University of Basel, Basel, Switzerland, **6** Disease Vector Group, Department of Plant Protection Biology, Swedish University of Agricultural Sciences, Alnarp, Sweden, **7** School of Public Health, University of Witwatersrand, Johannesburg, South Africa, **8** Institute of Biodiversity, Animal Health and Comparative Medicine, University of Glasgow, Glasgow, United Kingdom, **9** Department of Vector Biology, Liverpool School of Tropical Medicine, Liverpool, United Kingdom, **10** Department of Veterinary and Animal Sciences, Faculty of Health and Medical Sciences, University of Copenhagen, Frederiksberg, Denmark, **11** FaunaPhotonics APS, Copenhagen N, Denmark, **12** Norsk Elektro Optikk AS, Skedsmokorset, Norway

\* samuel.jansson@forbrf.lth.se

**Data Availability Statement:** The dataset presented in the manuscript is available at https://doi.org/10.6084/m9.figshare.13318454.v1.

## Abstract

Lack of tools for detailed, real-time observation of mosquito behavior with high spatio-temporal resolution limits progress towards improved malaria vector control. We deployed a high-resolution entomological lidar to monitor a half-kilometer static transect positioned over rice fields outside a Tanzanian village. A quarter of a million *in situ* insect observations were classified, and several insect taxa were identified based on their modulation signatures. We observed distinct range distributions of male and female mosquitoes in relation to the village periphery, and spatio-temporal behavioral features, such as swarming. Furthermore, we observed that the spatial distributions of males and females change independently of each other during the day, and were able to estimate the daily dispersal of mosquitoes towards and away from the village. The findings of this study demonstrate how lidar-based monitoring could dramatically improve our understanding of malaria vector ecology and control options.

## Introduction

Malaria is a predominantly tropical disease caused by *Plasmodium* parasites and transmitted by *Anopheles* mosquitoes, which still claims almost half a million lives each year and slows the economic development of the world's poorest countries [1–3]. Malaria risk is exacerbated by poverty and poor housing, especially in rural areas. Africa is disproportionately affected because it is home to several mosquito species that are exceptionally efficient vectors of the parasite because they specialize in feeding upon humans [4, 5]. Unprecedented reductions in malaria burden since the turn of the century have averted several million deaths, largely due to

**Funding:** SJ and MB were supported in part by Innovationsfonden, Denmark, by the Swedish Research Council through grants to Lund Laser Centre and the Centre for Animal Movement Research. MB was further supported by Lund University and by a direct grant from the vice chancellor. CK and MB are co-founders and shareholders but not employees of FaunaPhotonics, and per written agreement FaunaPhotonics had no influence on the scientific reporting. MB is an employee of Norsk Elektro Optikk, which is a non-profit company owned by a foundation with the aim to support optics and art in Norway. Norsk Elektro Optikk provided support in the form of salary for MB, but did not have any additional role in the study design, data collection and analysis, decision to publish, or preparation of the manuscript. The specific roles of MB are articulated in the 'author contributions' section.

**Competing interests:** CK and MB are co-founders and shareholders but not employees of FaunaPhotonics, and per written agreement FaunaPhotonics had no influence on the scientific reporting. MB is an employee of Norsk Elektro Optikk, which is a non-profit company owned by a foundation with the aim to support optics and art in Norway. This does not alter our adherence to PLOS ONE policies on sharing data and materials.

the implementation of vector control with insecticide-treated nets and indoor residual spraying of insecticides [6, 7]. However, malaria control is now truly at a crossroad, as progress has recently stalled for two major reasons [1, 8]. First, behavioral evasiveness of mosquitoes defines fundamental biological limits to the effects of insecticide-treated bed nets and indoor residual spraying, because both approaches selectively target mosquitoes only when they feed and/or rest inside human dwellings [5]. Second, increasing physiological resistance of mosquitoes to insecticides contributes to rebounding transmission [9, 10]. Further progress towards malaria elimination will undoubtedly require new technologies that target other vector behaviors [11, 12], notably those that occur outdoors and are widely distributed across landscapes. To this end, greatly improved understanding of the landscape ecology and baseline behavior of mosquito populations is required, so that the design and deployment of these new tools may be rationally optimized [13]. However, detecting and quantifying wild mosquito activities *in situ*, and mapping their distribution across landscapes remains a challenge [14–16].

In this study, we demonstrate the applicability of lidar (laser radar) for mosquito surveillance [17], by real-time *in situ* spatial profiling of malaria vectors, through the classification by their wing-beat modulation, at the periphery of an African village. We present data collected continuously over three days during the dry season, with no precipitation and virtually no wind during recordings. Details such as male swarming and nocturnal host-seeking of female mosquitoes, which were previously impossible to observe and quantify, are elucidated. We demonstrate that groups of male and female mosquitoes appear at different distances from the village and at different times of the day, and measure mosquito fluxes towards and away from the village.

## Methods

### Entomological lidar

A static invisible near infrared laser beam was transmitted above adjacent fields of a village. Insects transiting the laser beam at different distances from the system backscattered light onto different sections of a linear sensor. Thus, insect activity was resolved in space and time as measurements are conducted. Additional information relating to the size, wingbeat frequency, heading and flight speed was obtained for each individual insect observation based on the properties of the signal [18].

In this study, a 3.2 W 808 nm laser diode with vertical linear polarization was expanded with a refractor telescope (f600 mm, ø127 mm) and focused into a 2.5x23.3 cm (height by width) line at a remote neoprene termination target. Backscattered light was collected by a Newtonian reflector telescope (f800 mm, ø200 mm), transmitted through a 10 nm FWHM bandpass filter centered at 808 nm, and focused onto a 2048 pixel CMOS linescan camera. Transmitter-receiver separation distance was 814 mm, the camera was tilted 45˚ relative to the receiver telescope and the expander telescope was tilted roughly 1˚ relative to the receiver telescope, fulfilling the Scheimpflug condition [19]. An infinite focal depth was thereby achieved, with each pixel on the sensor sharply resolving a different section of the laser beam. The sensor line rate was 3.5 kHz, and the laser was turned on and off intermittently between exposures to enable background subtraction and daytime operation. A schematic of the system is shown in Fig 1.

### Field campaign

Lidar measurements were carried out continuously between August 31 and September 5, 2016, in the village of Lupiro, Tanzania. Ethical approval for the study was obtained from Ifakara Health Institute IRB (IHI/IRB/No: 34–2014) and Medical Research Coordination Committee

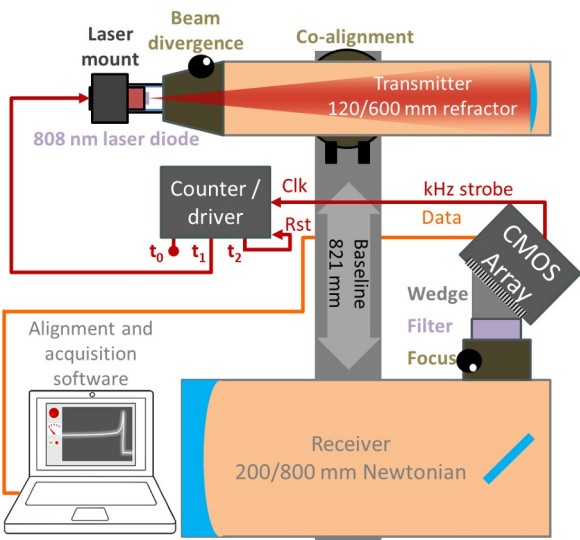

**Fig 1. Schematic of the lidar system used in Lupiro.** 808 nm light from a laser diode is expanded into a 102 mm diameter beam and transmitted through the air. The beam is terminated in a neoprene target attached to a tree 598 m from the lidar system. Backscattered light from organisms transiting the laser beam is collected by a Newtonian telescope and focused onto a detector array. The system has infinite focal depth, and due to the geometry of the configuration, light scattered from different sections of the laser beam is focused sharply onto different sections of the detector. The laser driver is used to intermittently turn the laser on and off, enabling the real-time acquisition of the optical background.

of the National Institute of Medical Research (Certificate No. NIMR/HQ/R.8a/Vol.IX/1903). The lidar system was positioned in a hut at the outskirts of the village (8°23'03.8"S, 36° 40'26.7"E) and powered with a 2 kW portable generator. The laser beam was transmitted in a roughly north-eastern direction, propagating 3–5 m above fields of corn and rice, and was terminated on a neoprene target attached to a tree 598 m from the lidar system (8°22'44.8"S, 36° 40'31.4"E). The probe volume consisted of the overlap between the laser beam and the field of view of the sensor. With the used laser, sensor and telescopes, the probe volume was 12 cm tall and 0.75 cm wide at 35 m (the near limit of the system), and 2.5 cm tall and 18 cm wide at 598 m, yielding a total probe volume of ~2 m$^3$. This orientation of the system is advantageous because the vertically linearly polarized light may impinge on insect wings at Brewster angle during wing beats, which may produce more detailed wave forms. It was also selected because a higher probe volume at close range may lead to a larger number of observed insects, whereas a wider probe volume at long range may lead to longer insect observations far away, resulting in better frequency resolution. The measurement site and geometry is shown in Fig 2. Mosquitoes were captured with a CDC light trap near the lidar system and species classified, see Table 1, enabling educated guesses on species identities of lidar-observed mosquitoes.

Weather data were collected concurrently with the lidar measurements using a USB weather station. Temperature peaks of 30–32°C were obtained together with the lowest relative humidity of about 40% in the afternoons around 15:00. The lowest temperatures and relative humidity peaks of 22–24°C and 70%, respectively, were obtained in the early mornings around 06:00. The wind speed peaked at 2.5 m/s at 10:00 September 4, but was below 1 m/s most of the time.

## Extraction and calibration of insect observations

The data was stored in binary files of 2048x35,000 16-bit data points, corresponding to 10 seconds of measurements. Every second exposure was respectively bright and dark,

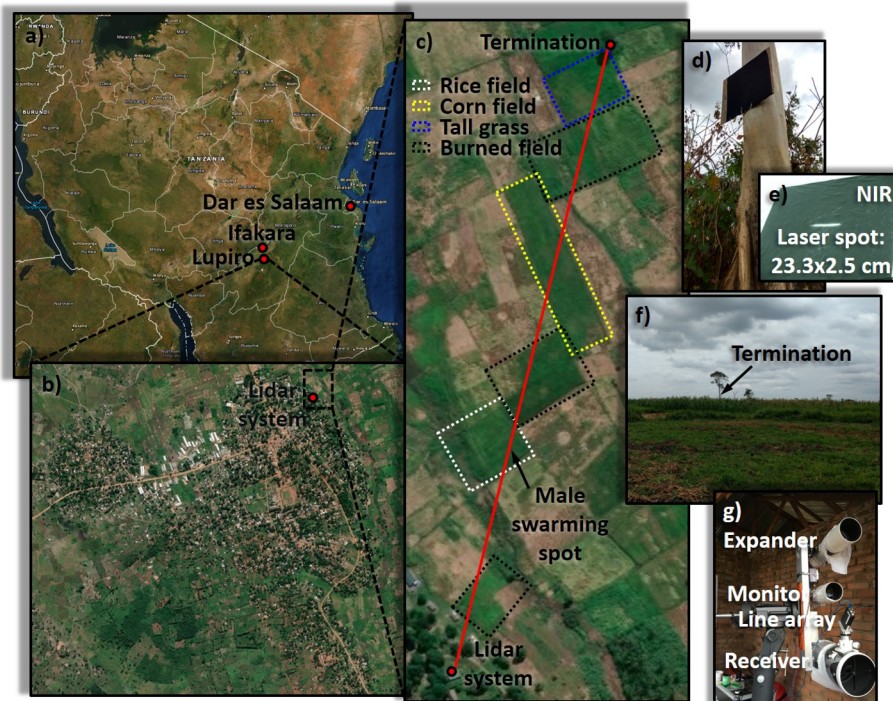

**Fig 2. Overview of the measurement location. a)** Satellite image of Tanzania showing the location of Lupiro village, in which the experiment was conducted. **b)** Satellite image of Lupiro village. The lidar system was located in the northeastern outskirts of the village. **c)** Satellite image of the measurement site. The laser beam was transmitted across the landscape, and was terminated at a distance of 598 m. The landscape contained fields of different crops, as well as empty and burned patches. **d)** Image of the neoprene termination target, mounted ~5 m above ground on a tree trunk. **e)** Near infrared photo of the termination target, showing the dimensions of the laser spot at that location. **f)** Photo of the termination target as seen from afar, giving an indication of the landscape and measurement conditions. **g)** Photo of the lidar system. The laser is transmitted through the expander telescope, and backscattered light is collected by the receiver telescope and focused onto a line array. A camera is connected to the monitor telescope, allowing the operator to aim the laser beam onto the termination target and giving a real-time overview of the experiment. Satellite images were obtained from Landsat, courtesy of the U.S. Geological Survey.

corresponding to the laser being on and off. The optical background in each pixel at each point in time was obtained through interpolation of the dark time slots and subtracted. A detection threshold with a signal-to-noise ratio SNR of 5:1 was set in each pixel as the median signal of the pixel plus five times the interquartile range (IQR). A binary map of all intensities exceeding the threshold was obtained and refined through image erosion and dilation. We obtained 456,721 data segments of high intensity, corresponding to insects transiting the laser beam, which were extracted from the raw data. The time duration of insect signals relates directly to which frequencies can be observed in the signals. The observable frequency range

**Table 1. Mosquitoes captured with a CDC light trap near the lidar.** A CDC light trap was placed in the village near the lidar system. Captured mosquitoes were species classified for comparison with lidar data.

| | Species | | | | | |
|---|---|---|---|---|---|---|
| **Date** | ***An. gambiae s.l.*** | ***An. funestus*** | ***An. coustani*** | ***Culex s.p.p.*** | ***Mansonia s.p.p.*** | ***Coquilettidia s.p.p.*** |
| 02-Sep-2016 | 536 | 5 | 5 | 161 | 8 | 0 |
| 03-Sep-2016 | 152 | 1 | 0 | 74 | 1 | 0 |
| 04-Sep-2016 | 482 | 1 | 1 | 279 | 11 | 3 |
| Total | 1170 | 7 | 6 | 514 | 20 | 3 |
| Proportion | 68,0% | 0,4% | 0,3% | 29,9% | 1,2% | 0,2% |

extends from the inverse of the time duration of a signal up to the Nyquist frequency, which is half of the sample rate. The minimum time duration of signals also determines the frequency resolution. To obtain a sufficient frequency resolution, 223,061 insect observations were discarded since their short transit times did not allow modulation spectra estimation, leaving 233,660 for further analysis. The full dataset is accessible at https://doi.org/10.6084/m9.figshare.13318454.v1.

The optical cross section (OCS) of the termination was calculated from the laser spot height (2.5 cm), the width of the probe volume and the reflectance of the neoprene termination target (1.8%). The signal across the entire range was calibrated into OCS through the inverse-square law and comparison to the integrated termination intensity. A time series $\sigma_{bs}$ for each insect observation was obtained by summing the extracted data segment along the range axis [20]. The parameterization process is explained in more detail in Malmqvist *et al* [18], and the steps are shown in Fig 3. However, the frequency analysis used here differs from our previous work, and is thus detailed below.

## Frequency parameterization

The backscatter signals from flying insects are modulated due to the insect wing beats. Wing-beat frequency is a good indicator of insect species, in particular for mosquitoes due to their characteristically high frequencies [21, 22]. However, accurately and robustly estimating the fundamental frequency of 233,660 time-series signal segments of varying duration and quality is a challenging task. Two methods were developed to tackle this problem [23, 24], and are explained below.

An insect signal can be divided into two components: the body signal, proportional to an envelope for the entire signal as the insect enters and exits the beam, and an oscillatory component due to the wing beat dynamics. In order to distinguish these two signal components, the WBF needs to be determined. A set of 500 test frequencies $f_{test}$ between the lowest observable frequency, defined by the transit time, and the Nyquist frequency, 875 Hz, was defined. For each insect observation, all test frequencies between the lowest observable and the Nyquist frequency were tested. A discrete time window was defined by the period time of the test frequency, and the signal envelope was acquired by taking the average of a sliding minimum- and a sliding maximum filtered signal. A discrete harmonic model containing the envelope and the sine- and cosine components of the test frequency and its overtones up to the Nyquist frequency was implemented. Furthermore, the frequency components were weighted by the envelope. The coefficients of the model were obtained through regression, and the root-mean-square error (RMSE) was calculated. Thereby, the RMSE of all test frequencies were obtained, yielding the error vector $e_{init}$.

This model is biased toward both very low and very high frequencies. At low test frequencies the model contains many overtones and degrees of freedom, yielding a lower RMSE (regressor bias). At high test frequencies, the time window used by the sliding minimum- and maximum filters is smaller, causing the envelope to explain both body and wing contributions (window bias). This means that the central frequency region in which most insect WBFs are found is the least likely to perform well in the model, and the residual $e_{init}$ needs to be adjusted for the biases to identify an unbiased WBF. This can be understood as punishing for information fed to the model through degrees of freedoms, either in the regressor or in the envelope time vector. The reasoning is similar to Akaikes criterions in information theory. The two biases to the frequency selection were treated separately. The regressor error was modelled analytically according to Eq 1,

$$\hat{e}_{reg} = 1 - N_{dof}/l, \tag{1}$$

where $N_{dof}$ is the frequency-dependent number of degrees of freedom of the model, and $l$ is the number of samples of the insect observation. A similar approach to modelling the window error

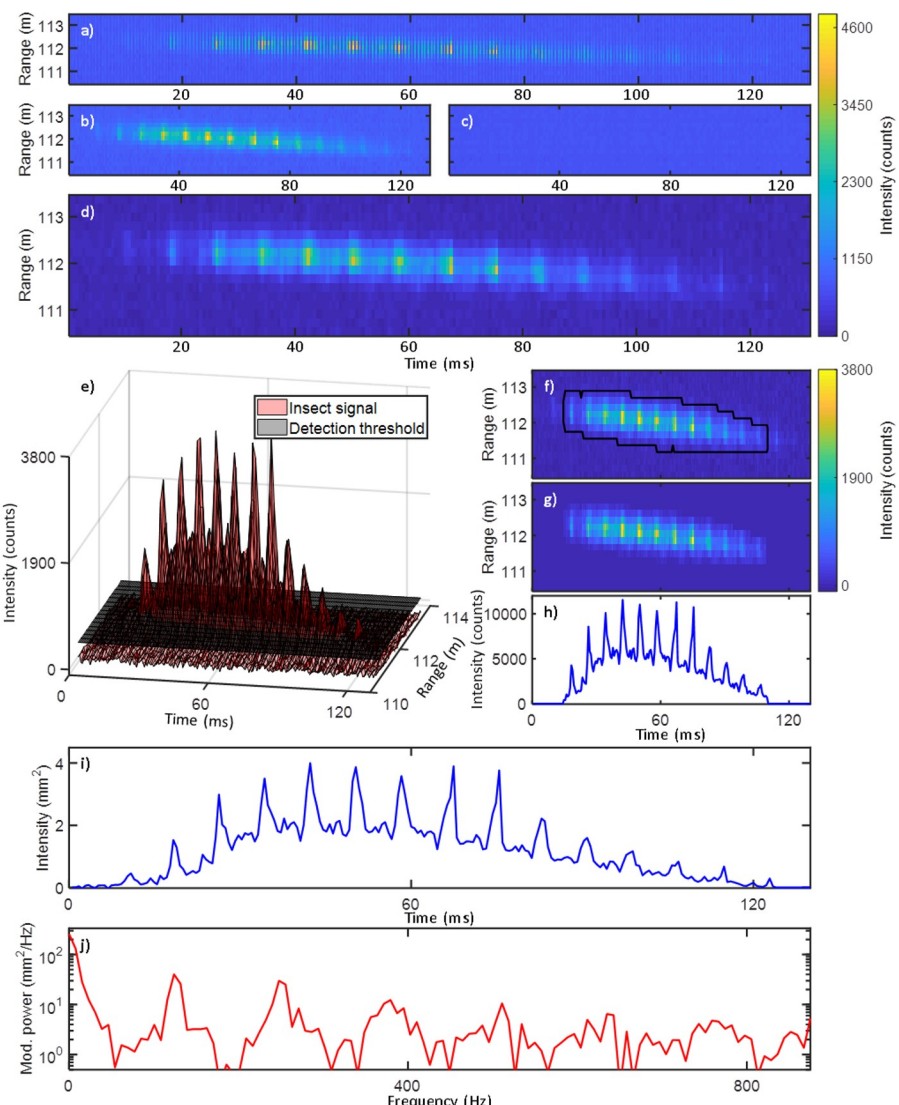

**Fig 3. Illustration of the data analysis procedure. a**) Raw data, in which every second exposure corresponds to when the laser is on and off, respectively. **b-c**) The raw data is sorted into the on- (**b**) and off components (**c**). **d**) The optical background is acquired from (**c**) through interpolation, and subtracted from (**b**). **e**) A detection threshold with an SNR = 2 is generated. A detection mask is generated to map all data segments which exceed the threshold. **f**) The detection mask (black line) indicates all instances of the signal exceeding the threshold. Image erosion and dilation are used to adjust the detection mask, filtering out signal segments too short to be of interest. **g**) The detection mask is used to crop out signal regions of interest. **h**) The signal is summed along the range axis, generating a time series. **i**) The signal intensity is calibrated into an optical cross section. **j**) Power spectrum of the time series in (**i**), with peaks at the insect wing-beat frequency and its overtones.

was attempted but found insufficient. However, since the window error $e_{win}$ is independent of the WBF $f_0$ it could instead be measured directly as the RMSE of the envelope and the insect signal. The product of $\hat{e}_{reg}$ and $e_{win}$ thus contains information on the frequency biases of the model, without being affected by $f_0$. The adjusted RMSE vector is obtained according to Eq 2.

$$\hat{e}_{final} = e_{init}/(\hat{e}_{reg}*e_{win}),\qquad(2)$$

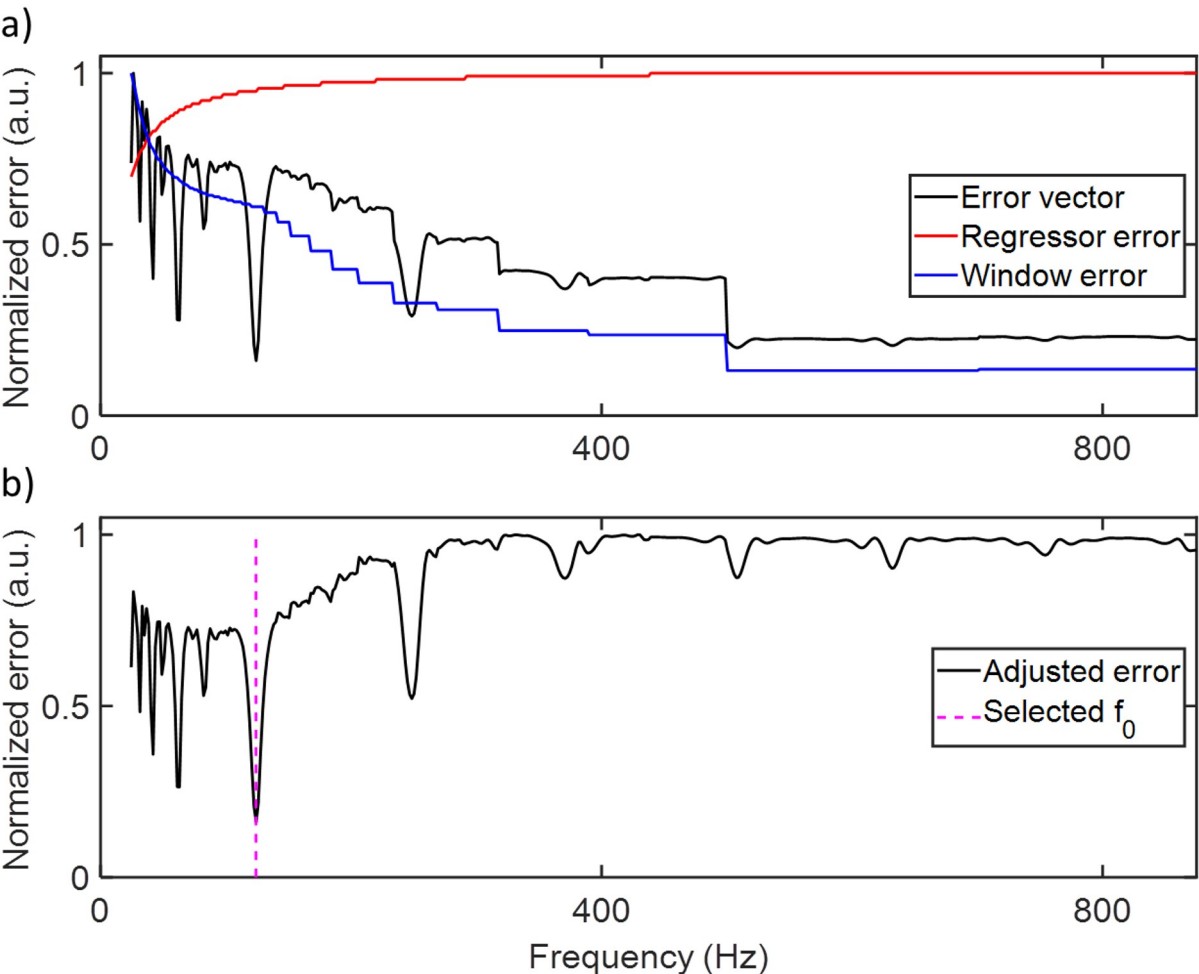

**Fig 4. a**) Initial error vector, regressor error and window error as function of test frequency. The regressor error indicates that there is a bias toward lower frequencies, and the window error indicates that there is a bias toward higher frequencies. **b**) The adjusted error vector. By eliminating the biases inherent to the model, the wing-beat frequency f0 can be selected with much improved accuracy.

$f_0$ corresponds to the minimum of $\hat{e}_{final}$. Fig 4 shows $e_{init}$, $\hat{e}_{reg}$, $e_{win}$ and $\hat{e}_{final}$ as function of $f_{test}$ for the same insect observation as shown in Fig 3, and marks the obtained WBF $f_0$.

Upon determination of $f_0$, insect observations could be further parameterized. A sliding minimum filter, with a window size equal to the period of $f_0$, was used to separate the signal backscattered by the insect body from that of the wings. Thus, the OCS of the bodies and wings of all insect observations were obtained. Additionally, the coefficients from the Fourier series model were used to calculate the strength and phase of $f_0$ and all overtones, thereby decomposing $\sigma_{bs}$ into a discrete set of components. Fig 5 shows the original signal $\sigma_{bs}$ together with the reconstructed signal from the Fourier series model, with wing- and body OCS marked.

## Hierarchical clustering

Due to the challenge involved in unbiased fundamental frequency estimation (well-known pitch detection problem, e.g. in speech and music recognition), an alternative approach was implemented. For each insect time series, the modulation power was calculated on a frequency

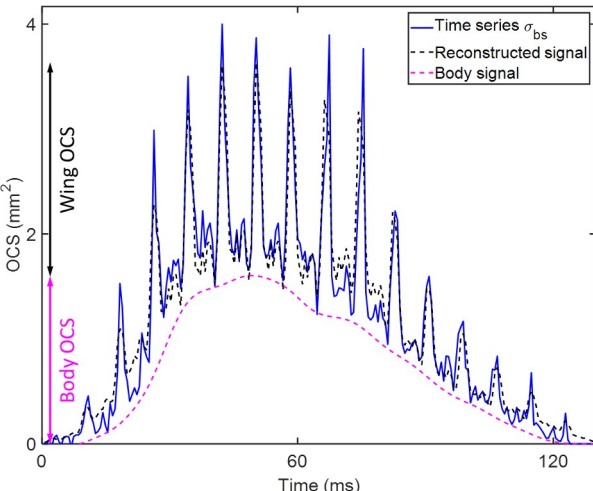

**Fig 5. The original and reconstructed insect time series.** The original time series is acquired from the data and calibrated. The wing-beat frequency f0 is obtained, after which the signal is reconstructed with a Fourier series containing the signal envelope and the sine- and cosine components of f0 and its overtones up to the Nyquist frequency. The body signal is obtained by applying a sliding minimum filter to the signal with a window size equal to the period of f0. The body- and wing OCS can then be acquired as the maximum of the body- and wing components of the signal, respectively.

scale with 40 equidistant bins between 85 and 875 Hz by Welch method (80% overlap, Gaussian window). The 40 frequency bins correspond to a 23 ms time window, which was the mode of all observed insect transit times. An insect power spectrum was thus obtained. A corresponding noise power spectrum was similarly obtained using a noise time series acquired at the same distance and within a fraction of a second of the insect signal. A linear regression model was applied to the noise spectrum. The insect power spectrum was divided with the regression model and subsequently normalized. All 233,660 normalized spectra were sorted into 20 clusters. This was done by calculating the Euclidean distances between all pairs of observations, which is a multi-dimensional expansion of the Pythagorean theorem, and grouping similar observations (i.e. with short Euclidean distances) together. The clusters were labeled according to their frequency contents based on literature values [21, 22, 25].

## Cluster and frequency interpretation

Male and female mosquitoes were differentiated from other insects by their modulation signatures and high pitch. Clusters with $f_0 >= 550$ Hz correspond to male mosquitoes, clusters with 300 Hz $<= f_0 < 550$ Hz correspond to female mosquitoes [22], clusters with $f_0 < 300$ Hz correspond to other insects, and clusters with high-intensity signals correspond to larger insects. Clusters lacking a distinguishable wing-beat frequency were labeled as unknown were excluded from further analysis. Fig 6 shows a dendrogram and the average spectrum and variance of all clusters. The labels and number of observations of each cluster are also indicated. Some further comparisons of signal parameters between clusters were made. Fig 7 shows histograms of the maximum OCS and transit time $\Delta t$ of all labeled clusters. As a general trend, mosquito clusters display the lowest OCS values out of the groups, which is consistent with their size. Low-frequency insects display slightly higher values, and clusters labeled as larger organisms display the highest values. Mosquitoes and low-frequency insects display similar transit times, whereas larger organisms display shorter transit times that could correspond to higher flight speeds. Cluster 2 displays the highest modulation frequency of the female clusters, and is

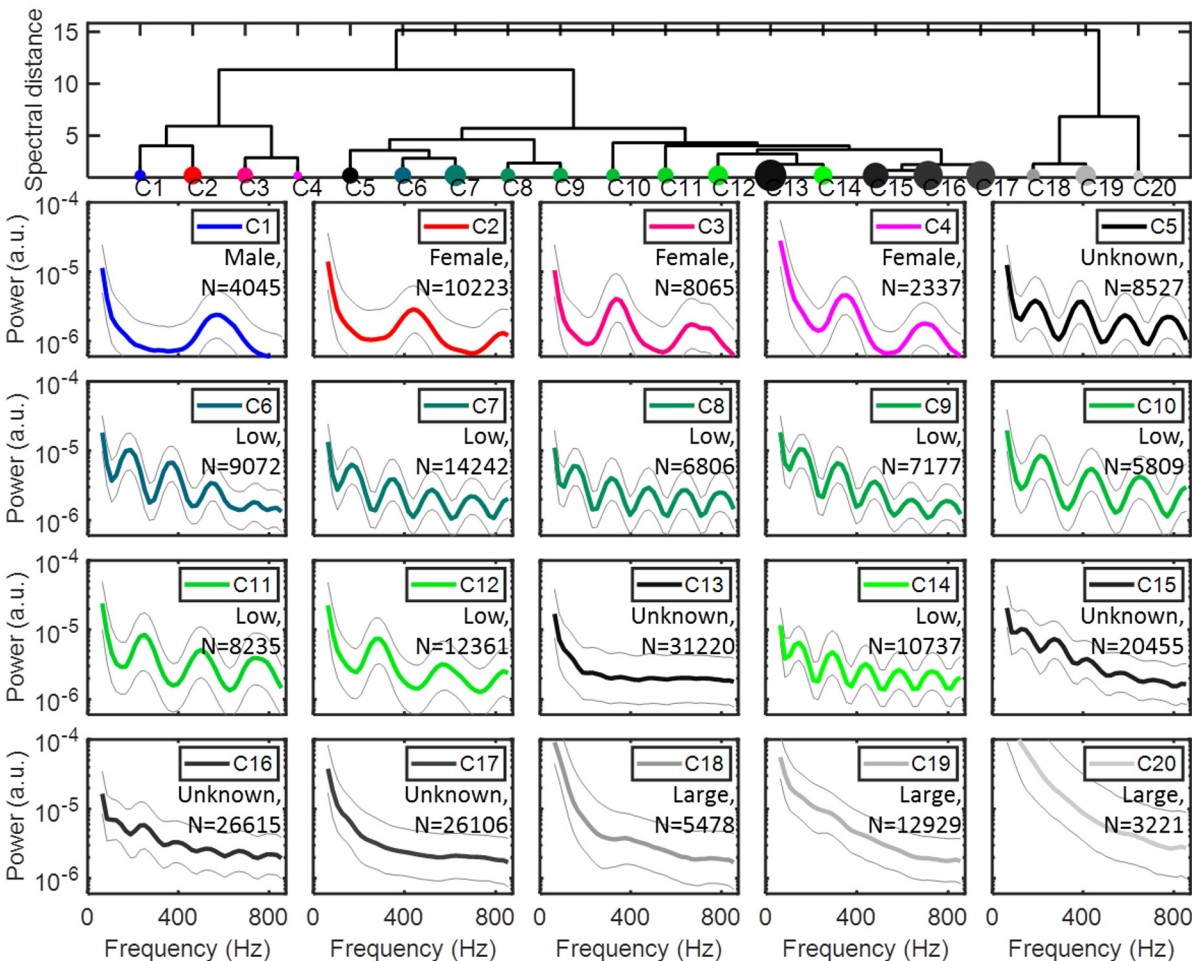

**Fig 6. Illustration of the 20 obtained clusters.** The dendrogram at the top shows how closely related the different clusters are, with three distinct groups of clusters emerging. Clusters 1–4 correspond to mosquitoes based on their high frequency contents. Clusters 5–17 contain a mix of low-frequency insects and unclassifiable observations, whereas clusters 18–20 contain observations corresponding to larger insects or vertebrates. The average spectrum, as well as minimum and maximum, are shown together with the number of observations in each cluster.

henceforth labeled high-frequency females. Clusters 3 and 4 display very similar modulation frequencies, and likely belong to the same species. Cluster 3 displays longer transit times and lower OCS values, whereas cluster 4 exhibits higher OCS values and shorter transit times. This indicates that C4 mosquitoes transit the probe volume laterally, whereas C3 mosquitoes fly more along the laser beam.

## Results and discussion

Entomological lidar measurements were carried out in the village of Lupiro in southern Tanzania (Figs 1 and 2). A near-infrared (NIR) diode laser was transmitted horizontally across cultivated fields and terminated in a distant target. Data was collected continuously for a period of 3 days (September 2 to 4, 2016). We analyzed 233,660 insect observations and obtained their optical cross sections (OCS), wing-beat frequencies (WBF) and power spectra (Figs 3–5).

Insect observations were hierarchically clustered based on the Euclidean distance between their power spectra, and the first 20 branches of the dendrogram were interpreted. Based on the centroid frequency contents [22, 25–28], clusters were labelled as 'male mosquitoes',

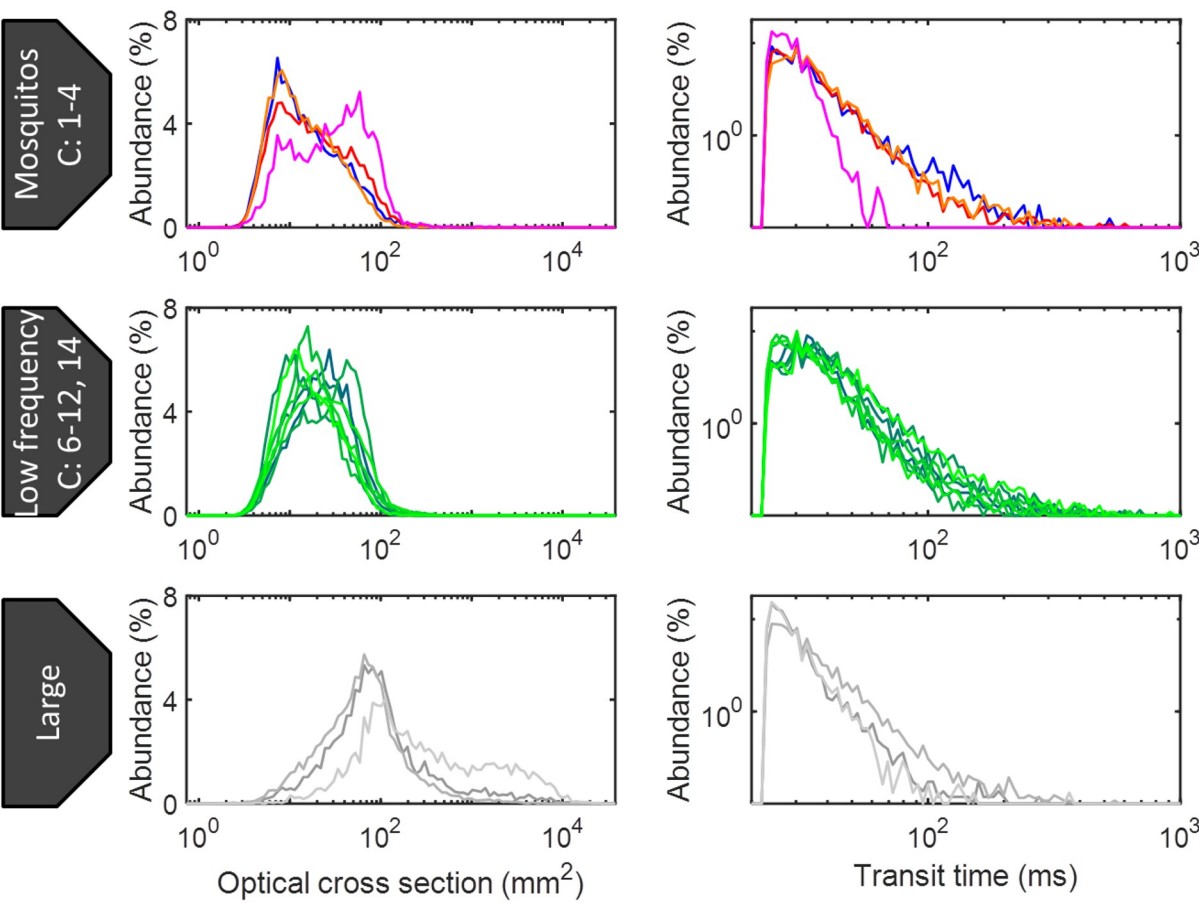

**Fig 7. Cluster interpretation.** Histograms of optical cross section (left) and transit time (right) for the observations labelled as mosquitoes (top), low-frequency insects (middle) and larger organisms (bottom). The optical cross section of mosquitoes is generally lower than that of the other groups. The transit time of the large observations is shorter, which could be due to a higher flight speed. Cluster 4 (female mosquito) displays a higher optical cross section and a shorter transit time than the others, which could indicate a lateral transit of the laser beam.

'female mosquitoes', 'low-frequency insects', 'large organisms' or 'unknown' (Fig 6). In the subsequent analysis, three overarching groups of insects were considered: male mosquitoes (one cluster), female mosquitoes (three clusters) and other insects (eight clusters corresponding to the 'low-frequency insects'). We observed 2,698 male mosquitoes, 13,820 female mosquitoes and 55,006 other insects during the measurement period, and their distribution in space and time was investigated. The overall range distribution of male and female mosquitoes as well as other insects is shown in Fig 8, and the 2-Dimensional time-range histograms of the three groups are shown together with their WBF distributions in Fig 9.

The decrease in insect counts with range seen in Fig 8 is a product of the insect distribution and instrument sensitivity [29]. Throughout the study period, mosquitoes were observed closer to the village than other insects, and males were observed closer to the village than the females. However, since the distributions are largely attributed to the system sensitivity, they were nevertheless more alike than dissimilar. Large and small insects were affected differently, thus making comparisons between insect groups challenging. The distributions can be approximated with a power law, $N = N_0 \cdot r^{\alpha}$, in which the range decay exponent $\alpha$ sheds light on group-specific range dependencies.

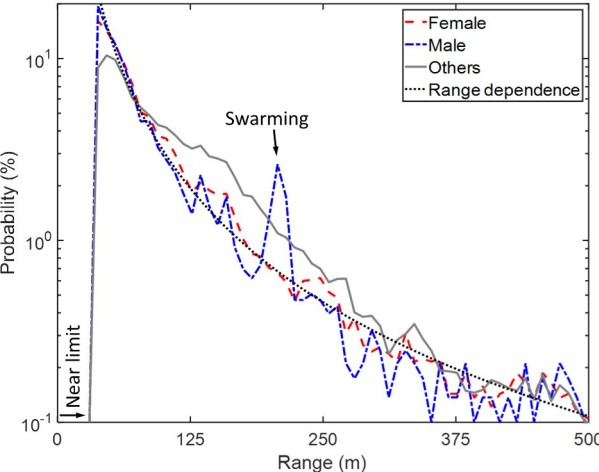

**Fig 8. Distributions of range (detection distance) of female and male mosquitoes compared to other insects.** The distributions contain data from three full days. The near limit is the minimum distance at which organisms are detectable, corresponding to the distance at which the laser beam and sensor field of view start overlapping. The reduced instrument sensitivity with range yields a decreasing probability of detection with increasing range for all insect groups. Mosquitoes were detected closer to the village on average than other insects (median distances: 65 m and 62 m for female and male mosquitoes, versus 86 m for others), and narrow-range features such as the swarming of males are shown. Different insects display distinct features in their range distributions from the village, suggesting aggregation of mosquitoes around the village when compared to other insects. This can in part be attributed to the sensitivity decrease with detection distance, and mosquitoes being smaller than other insects. The range dependence of the distributions can be approximated with a power law, $N = N_0 \cdot r^\alpha$.

The WBF distributions of mosquitoes in Fig 9 coincide with corresponding distributions previously described [22, 30], as well as with the fundamental frequencies of the corresponding clusters (Fig 6). This serves as a complement to the clustering method, independently indicating that the cluster interpretation was correct. The majority of insect activity takes place just before dawn and right after dusk (Fig 9), consistent with previous studies [31, 32]. The activity of female mosquitoes after midnight near the village was observed more frequently compared to that of males (compare Fig 9A and 9B), i.e. during the peak biting activity period for anthropophagic malaria vectors, such as *Anopheles funestus* [32]. Compared to female mosquito activity during the rest of the day, females at night time exhibit longer transit times and smaller cross sections (Fig 10). This shows that the mosquitoes are flying along the beam, toward or away from the village rather than parallel to the village border, indicating that they may be actively seeking a blood meal or have successfully obtained one.

Swarms of males were observed, spatiotemporally confined within a distance of ~210 m from the lidar at 18:45 in the evenings, 13 min post sunset. Repeated observations of male swarms during three consecutive nights were made (Fig 11), with the swarms appearing at the same minute in the same location each night and remaining in the beam for 3 min. The spatial extension of the swarm reads 17 m, but is due to the range uncertainty at the distance of the swarm [33]. The swarm location coincides with a foot path through a rice plantation (Figs 2 and 12), which has previously been identified as a common swarming spot for male *An. funestus* and *An. arabiensis* mosquitoes [34, 35]. A total of 16 female mosquitoes were observed entering the swarms of males (Fig 11), likely *Anopheles spp.* based on their wing-beat frequencies.

As shown in Fig 6, three clusters of insects were interpreted as female mosquitoes. Based on the characteristics of the three clusters, these are labelled as high-frequency females (C2), parallel females (C3) and perpendicular females (C4). High-frequency females exhibit high WBFs,

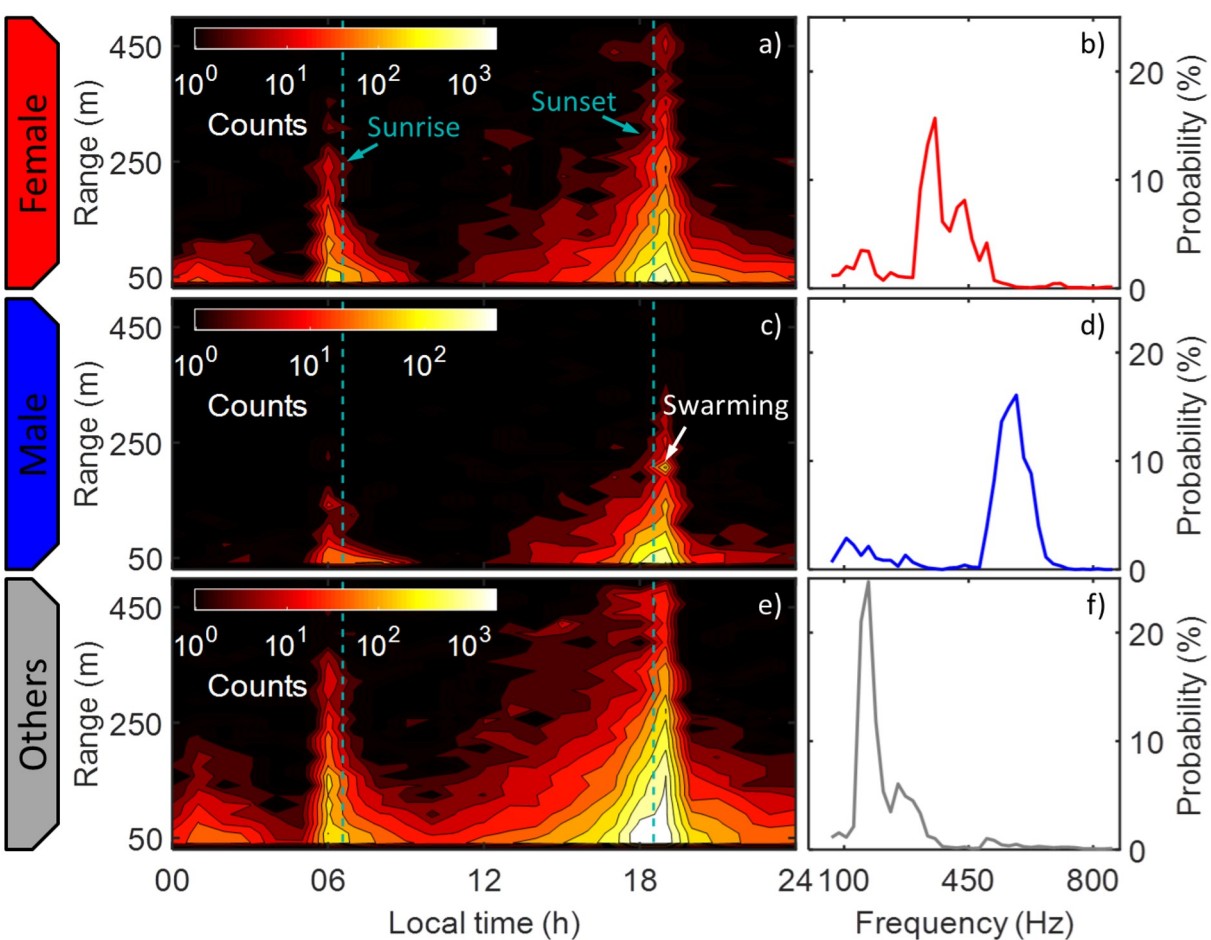

**Fig 9. Activity of female and male mosquitoes, as well as other insects throughout the day. a,c,e)** Smoothed contours of 2D histograms (1 h centered on the hour x 16 m bins) of the detected counts during 3 days with logarithmic color coding. Interesting features include the crepuscular activity peaks at dawn and dusk, the male swarming at 210 m and the nightly female activity. The classes are based on hierarchical clustering. **b,d,f)** Distributions of fundamental wing-beat frequencies of the observed insects in each group, estimated with an independent method [23].

whereas parallel and perpendicular females exhibit lower WBFs split into two separate clusters with differing body/wing ratios, indicative of heading in different directions. Parallel females fly along the laser beam, toward or away from the village, whereas perpendicular females fly straight through the beam, parallel to the village perimeter delimited by flood-prone rice fields. Based on laboratory measurements [30], high-frequency females likely correspond to mixed *Culex spp.*, whereas parallel and perpendicular females appear more likely to be *Anopheles*. For more information, see cluster interpretation in methods and Fig 12. Fig 13 shows the activity per time of day of the different insect groups. The three clusters of female mosquitoes are shown separately for comparison. Prior to sunset, parallel females are the earliest to initiate activity. These females may correspond to unfed females, many of which could also be unmated and therefore seeking males [36], with low WBFs due to a lack of payload [37]. Males appear ~15 minutes after the parallel females, followed by high-frequency females that appear after another ~20 minutes. Perpendicular females are the last to become active, appearing ~15 minutes after high-frequency females, and do not come out in large numbers until the major evening peak at dusk. This is the least abundant female group, corresponding to roughly 25%

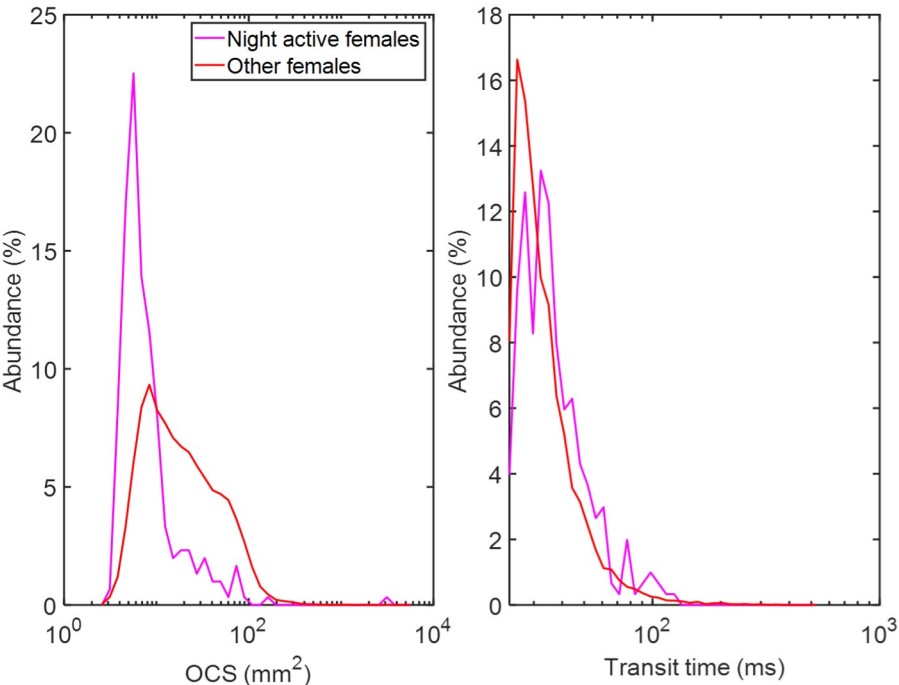

**Fig 10. Night-active females.** Histogram of optical cross section (left) and transit time (right) of night-active female mosquitoes and all other female mosquitoes. When an insect is observed from the front, flying along the laser beam, it appears small and stays in the beam for a comparatively long time. Adversely, when an insect is observed from the side, flying straight through the laser beam, it appears large and stays in the beam for a short time. As observed, night-active female mosquitoes are smaller and remain in the beam longer than other females. This indicates that they are flying along the beam, toward or away from the village, to a larger extent than other female mosquitoes.

of the parallel or high-frequency females. Males and parallel females peak in activity during the male swarming time at 18:45 in the evening. High-frequency females and other insects display peak activity slightly later, at 18:55, and perpendicular females peak in activity last at 19:00.

Fig 14 presents estimated fluxes to or from the village at six distinct time intervals, summarized over all three days and exhibiting different insect activities. A peak of activity occurs prior to sunrise (5:40–6:50), with some lingering activity post-sunrise (6:50–8:40), particularly among male mosquitoes. The activity during the day (8:40–17:00) is generally low, but increases gradually prior to sunset (17:00–18:20). The highest activity peak is observed post-sunset (18:20–19:40), and the activity then decreases to relatively low levels during the night (19:40–5:40). The activity peaks in the morning and evening are consistent with other studies, but the nightly activity is comparatively lower than those reported by others [37, 38]. This may be because the lidar transect was 3–5 m above ground, whereas most mosquito activity is thought to occur closer to ground [32].

As mentioned previously, the decreasing insect counts with distance from the village (Fig 8) can be approximated by a power law, $N = N_0 \cdot r^\alpha$. By comparing the range decay exponent $\alpha$ for an insect group at different times of the day, significant differences in the distributions can be observed. The range decay exponent was calculated for all groups of insects during the aforementioned time intervals. The power $\alpha$ is negative due to the decreased instrument sensitivity with distance, with high magnitude values corresponding to mosquitoes congregating closer to the village. The net flux of insects, i.e. the number of insects from each group flying outwards subtracted by the number of insects flying inwards, weighted by transit time for

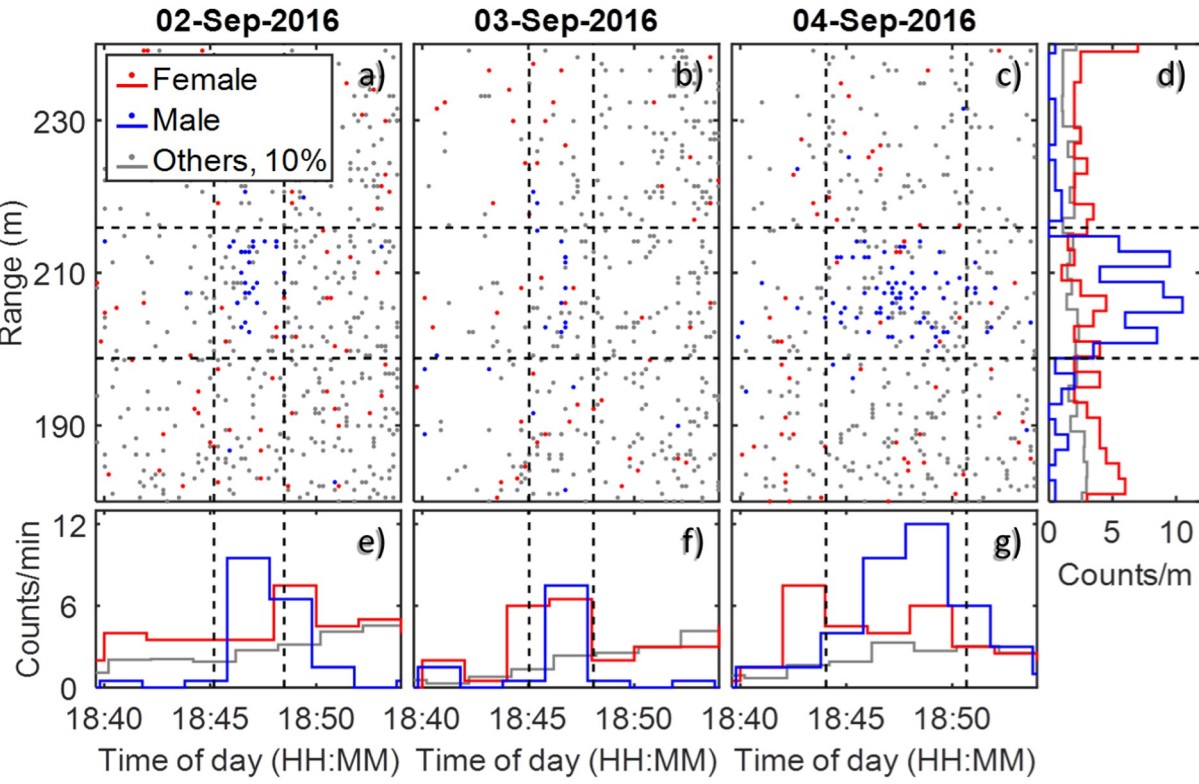

**Fig 11. Male mosquito swarming over three consecutive days.** The swarm boundaries, obtained as median ± interquartile range (iqr) of the distributions, are marked with dashed black lines. **a-c)** Time and range of each insect observation during the male swarm. **d)** Range histogram of the three groups of insects. The counts of "other" insects are reduced by 90% for comparison. Female mosquitoes and "others" exhibit flatter range distributions, whereas male mosquitoes are highly localized around 210 m from the village. **e-g)** Time histograms of the three insect groups during the male swarm. The swarming takes place early during the dusk activity peak, and rising flanks are observed among the female mosquitoes and other insects. In contrast, male mosquitoes exhibit a sharp peak during the swarm, and then dwindle quickly in numbers.

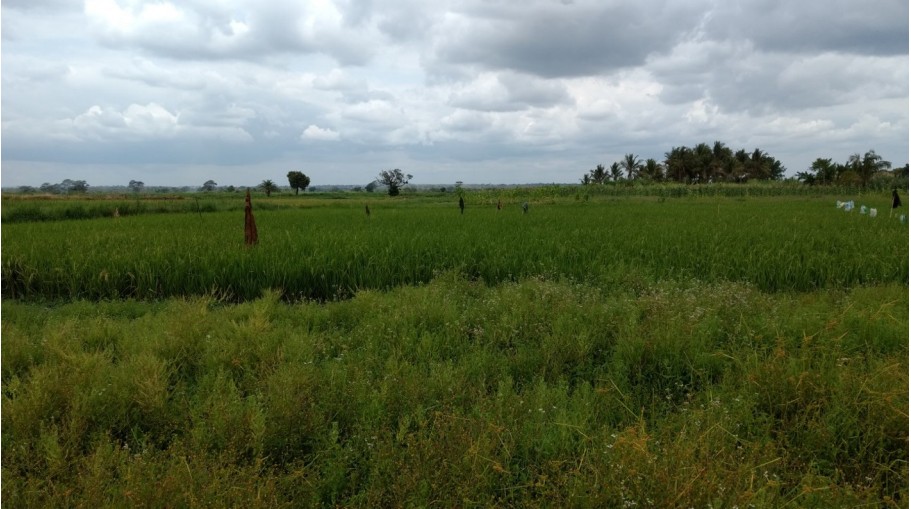

**Fig 12. Photo of the rice field and foot path where the swarms of male mosquitoes were observed.** The field is marked in Fig 2, and the photo is taken from the adjacent field SSE of the rice. The foot path was one of the larger ones in the area, and was commonly used by workers going to and from the fields in the mornings and evenings.

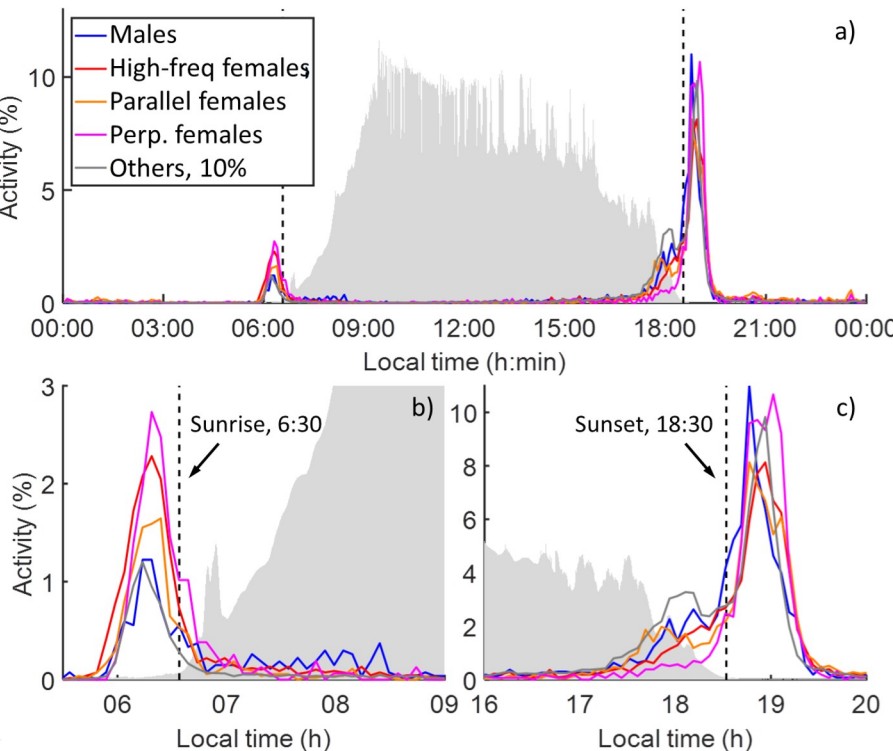

**Fig 13. Diurnal activity of the different insect groups. a)** Histograms of the diurnal activity of the observed insects (5 min bins), accumulated over the entire measurement period. The background light level is indicated in light grey. The nocturnal peak observed in logarithmic scale in **2a** is too small to appear in linear scale here. **b)** Close-up of the dawn activity peak. Male mosquitoes display lower activity during the initial rush, but linger for several hours afterwards. This post-dawn activity is consistent through all days. **c)** Close-up of the dusk activity peak. Different groups display peak activities, at different times.

improved accuracy, was calculated and is shown together with $\alpha$ in Fig 14. The confidence interval of $\alpha$ reflects how well the insect distribution is represented by the power law. The confidence interval may therefore be small even when there are low insect counts, as for male mosquitoes during the day time.

Insects are observed close to the village after dawn and during night, and further from the village before dawn, during the day and after dusk (Fig 14A). Note that the spatial distributions change significantly during the day, and the changes are distinct among the various groups. The majority of insect flux occurs around sunset, going in towards the village. During the rest of the day, the net flux is generally aligned outwards, away from the village. Whereas there is a strong incentive for host-seeking females to disperse towards the village, the efflux may be less directed as mosquitoes move away to oviposit because the village was surrounded on that side by suitable breeding sites. Although studies using methods such as human landing catch (HLC) have shown that most of the measurable biting occurs at night [39], the crepuscular dispersal activity of mosquitoes demonstrated here is consistent with field studies carried out elsewhere with vehicle-mounted sweep nets [40, 41]. In addition, simulation analyses suggest that HLCs may exaggerate measurements of feeding activity at times when most residents sleep under nets [42]. However, whereas HLC-based observations catch host-seeking individuals, lidar-observed mosquitoes are likely in different physiological states such as homing, mating or swarming, and therefore not directly comparable. To our knowledge this is the first study in which the dispersal direction has been investigated.

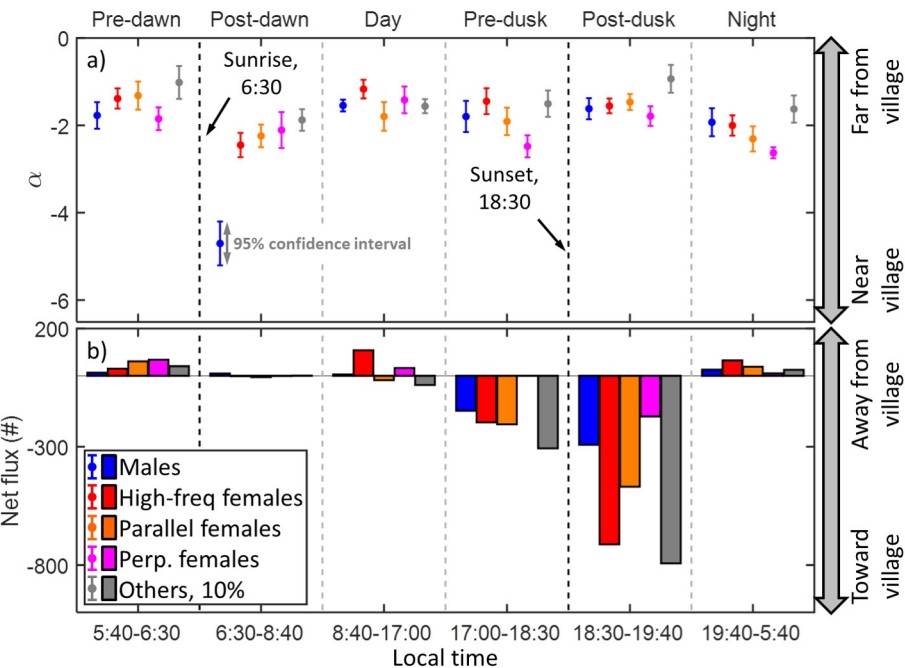

**Fig 14. The range decay exponent $\alpha$ and the net flux of insects across all three days of measurement. a)** The exponent $\alpha$ indicates how skewed insect distributions are towards the village. Negative values of greatest magnitude indicate close proximity to the village, whereas values closer to zero indicate that insects are detected further from the village. The fitted exponent α is presented with 95% confidence intervals. **b)** The net flux was calculated for all insect groups at different times of day. In the night, around dawn and during the day, the flux mostly goes away from the village. Around dusk, the flux mostly goes towards the village. The total counts differ between groups, which affects the fluxes. The counts of "other" insects are reduced 10-fold for ease of comparison.

Mosquitoes are observed closer to the village than other insects at all times, except for during the day. This fits well with the anthropophilic nature of African malaria vector mosquitoes. Male mosquitoes exhibit significant "lingering" activity after the dawn peak, unlike females and other insects (Fig 13B), which may be due to the different life requirements of the two sexes. Interestingly, this morning male activity is concentrated far closer to the village than the activity of all other groups at all times of the day (Fig 14A). This pattern is consistent across the first two days, but the male counts were too low in the third morning to discern whether or not it occurred then as well. We speculate that there may have been more nectar sources, resting places or females near the village at the time of the measurements, but are unable to verify this. At other times of the day, males were observed at intermediate distances from the village (Fig 14A). High-frequency females were observed near the village at night, far away during the peak before sunrise, and closer to the village afterwards. During the day and around sunset these females are observed far away. The nightly activity near the village may correspond to host-seeking females. After dawn, the activity may correspond to females with a heavy payload, observed close to the village just after taking a blood meal, consistent with previous simulations [42]. In that case, they could be looking for an oviposition site. Perhaps more likely, they may have been gravids that had rested while digesting and gestating until they ran out of time and dispersed at dawn. Before and after sunrise, parallel mosquitoes are observed farther and nearer, respectively, than their average. This female group was found at intermediate distances during the day and prior to sunset, slightly further away after sunset and relatively close at night. As this group is responsible for the increased nightly activity observed soon after midnight in Fig 9A, it may contain host-seeking *An. funestus* [39] although they can presently not

be distinguished from *An. arabiensis* in lidar data. Perpendicular females, being the least numerous female group, display overlapping distributions at intermediate distances throughout the morning and day. Before sunset and at night they are observed near the village, whereas during the activity peak after sunset they are observed at intermediate distance. Other insects are observed furthest away from the village during the activity peaks before sunrise and after sunset, and display overlapping distributions relatively far away at all other times. Applying our power law model to the data from another study [43] yielded $\alpha$ = -0.9±0.2, which is comparable to our results.

Weight and temperature are two factors affecting the WBFs of mosquitoes [28, 44, 45]. A female *An. arabiensis* weighs roughly 1.7 mg, a blood-fed female weighs approximately 3.8 mg and a gravid mosquito weighs 2.7 mg [44]. Mosquitoes feeding on nectar ingest about 0.38 mg [46], but may eat as much as a few mg when starved [47]. The weight gains correspond to frequency shifts of about 28% and 8.5% for blood-fed and gravid mosquitoes, respectively [45]. A 28% frequency shift is enough to confuse a female mosquito with a male one, but females are known to remain stationary while digesting blood meals. Thus, we expect this to have little effect on the results. An 8.5% frequency shift may cause confusion between the different groups of females, but is not significant enough to cause confusion between sexes. It is worth noting that the WBFs of perpendicular females match an 8.5% shifted WBF of parallel females very well. Perpendicular females may thus correspond to gravid parallel females. Regarding the temperature, the WBF is shifted about 2.8% per K [28], corresponding to an 11.2% difference between the morning and evening activity peaks (4 K difference). This is not significant enough to confuse sexes in the analyses, but may confuse parallel and perpendicular females. In particular, parallel females may be mistaken for perpendicular females in the early evening when the temperature is high. Since the perpendicular activity is very low prior to dusk, we conclude that this is unlikely. The weight gain from a typical nectar meal yields a smaller frequency shift than that of gravid mosquitoes, and is therefore unlikely to lead to misclassification. Large nectar meals as ingested by starved mosquitoes yield large frequency shifts that could lead to misclassification. However, as in the case with blood meals, mosquitoes tend to remain stationary while digesting these meals.

Perpendicular and parallel females displayed their primary influx towards the village after sunset, and efflux before dawn. Parallel females were active earlier than other females and were more night-active. They were active during male mating swarms and at night, and were generally flying along the beam, towards or away from the village. At night, they displayed noticeable activity near the village, and appeared further away and flying outwards before dawn. These observations indicate that the group may correspond to hungry and highly motivated females, in search of blood and/or a mate. Although the mating swarm of males we observed formed 210 m from the village, there may be many other swarms at different locations. Perpendicular females, which exhibited WBFs very similar to those expected from gravid parallel females, were generally flying laterally across the beam rather than along it. Out of all groups, their activity was the most concentrated to the crepuscular peaks, during which they were active almost exclusively before sunrise and after sunset. Should they correspond to gravids, flying in optimal conditions to avoid predators would make sense. Also, it would make sense that gravid mosquitoes which have been resting and waiting for the opportunity all day would begin dispersing en masse in the evening, whereas others with less-developed eggs may defer such activity until dawn and then choose between either dispersing before the heat of the day sets in or waiting it out until sunset. The less directional flight towards the village of perpendicular females is also consistent with the interpretation that these correspond to gravid mosquitoes, because they would be dispersing to larval habitats which were widely distributed in all directions eastward of the village. These two mosquito clusters closely match *An. arabiensis* in WBF

[22], and may thus correspond to gravid and host-seeking states of this species, which by far is the most abundant *Anopheles* species in this location and the only one from the *An. gambiae* complex. Based on the spike activity of parallel females after midnight, the group may also contain some *An. funestus* [48]. Like the other groups, high-frequency females displayed a very directed flux towards the village around sunset. As for the parallel and perpendicular females, the efflux of high-frequency females that took place during the rest of the day was less directed. As previously highlighted, this group displayed activity resembling that predicted for blood-fed or gravid females at night. Based on their WBFs, we expect that these correspond to *Culex* mosquitoes [30]. Since the hierarchical cluster analysis (HCA) yielded only one cluster of male mosquitoes, we conclude that this cluster likely contains both *Anopheles* and *Culex* males. Studies carried out in laboratory environments with a limited set of mosquito species generally report classification accuracies in the range of 70–90% [30, 49]. Misclassified abundant ones could therefore obscure a rare species. However, our trap catch in Table 1 contained 68% *Anopheles gambiae s.l.* and 29.9% *Culex spp*. mosquitoes. The remaining 2% can be assumed to have limited impact on the overall results.

## Conclusions and outlook

In this work, we demonstrated that modulation signatures obtained with lidar can be used to differentiate different types of insects, revealing behavioral patterns that were previously impossible to observe. In particular, we demonstrated that male and female mosquitoes can be distinguished in field conditions using lidar. Behaviors such as male swarming and the potential host-seeking of anthropophilic malaria vectors were elucidated. Females entering male swarms to mate were observed and may be studied in more detail with longer-running measurements and more intensive statistical analyses. We also showed that different groups of insects exhibit different activity levels throughout the day, and peak in activity at slightly different times. As demonstrated previously, this may be related to predation pressure [31]. Insects were also observed at different distances from the village at different times of day. We showed that the majority of insect influx towards the village occurred in the evenings, in relation to sunset, and that insects mostly disperse outwards, away from the village, during the rest of the day.

Future studies could be carried out in conjunction with vehicle-mounted sweep net drives, yielding an unbiased sample of the insect population for correlation with the lidar measurements. They could also benefit from *in-situ* characterization of optical properties and wing-beat harmonics of local insects. However, devices capable of such characterization are currently cumbersome and restricted to laboratory use, and further improvements are necessary. Recently developed line sensors with higher sample rates could be implemented in lidar systems, which would potentially improve the frequency analysis and classification. Additional spectral- and polarization bands have been shown to enable the classification of similar species [22] and the distinction of gravid from non-gravid females [50] in the laboratory, despite the overlapping WBF distributions of the groups. Radial activity maps could be obtained by scanning the laser beam slowly over a field. This may be used to indicate mosquito hot spots and improve collection strategies and the geopositioning of supplementary malaria vector control interventions such as attractive targeted sugar baits or odor-based traps.

## Acknowledgments

We appreciate the cross-validation of data and analysis by Jord Prangsma, Alfred Strand and Klaes Rydhmer, and we thank Flemming Rasmussen for assistance in the field. We thank Alexandra Andersson for her efforts with data analysis, and Alem Gebru for his work with optical

reference measurements. We acknowledge Anna Runemark, Maren Wellenreuther and Susanne Åkesson for general support and discussion.

## Author Contributions

**Conceptualization:** Samuel Jansson, Elin Malmqvist, Gerry Killeen, Mikkel Brydegaard.

**Data curation:** Samuel Jansson, Elin Malmqvist, Mikkel Brydegaard.

**Formal analysis:** Samuel Jansson, Elin Malmqvist, Mikkel Brydegaard.

**Investigation:** Samuel Jansson, Elin Malmqvist, Yeromin Mlacha, Mikkel Brydegaard.

**Methodology:** Samuel Jansson, Elin Malmqvist, Mikkel Brydegaard.

**Project administration:** Mikkel Brydegaard.

**Supervision:** Mikkel Brydegaard.

**Validation:** Samuel Jansson, Elin Malmqvist.

**Visualization:** Samuel Jansson, Elin Malmqvist.

**Writing – original draft:** Samuel Jansson.

**Writing – review & editing:** Samuel Jansson, Elin Malmqvist, Yeromin Mlacha, Rickard Ignell, Fredros Okumu, Gerry Killeen, Carsten Kirkeby, Mikkel Brydegaard.

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
