## [Decision Letter · Decision Letter 0]

4 Aug 2020

PONE-D-20-15981

Real-time dispersal of malaria vectors in rural Africa monitored with lidar

PLOS ONE

Dear Dr. Jansson,

Thank you for submitting your manuscript to PLOS ONE. After careful consideration, we feel that it has merit but does not fully meet PLOS ONE’s publication criteria as it currently stands. Therefore, we invite you to submit a revised version of the manuscript that addresses the points raised during the review process.

Two reviewers have shown strong support for this study and note how it represents an exciting advance in using lidar for the study of vector-borne diseases. Most comments are generally minor but will strengthen the manuscript, such as making the findings more accessible to a generalist audience and in particular those working on arthropod vectors themselves. Both reviewers also flagged some ambiguity about what vector species actually occur in this region of Tanzania; distributions of known mosquito species, or other ground-truthed information on vector species (including not only mosquitoes but also the "other" category of insects), would improve your interpretations. Some comments are also offered regarding how the discussion could describe how this method could improve sampling strategies or control as well as how it could be scaled up across geographies. *Plasmodium* should also be italicized.

From the editorial perspective, the raw data to reproduce the analyses are not included. Note that PLoS ONE requires such data be deposited in the supplemental material or an external repository for publication.

We look forward to receiving your revised manuscript.

Kind regards,

Daniel Becker

Academic Editor

PLOS ONE

Journal Requirements:

We note that one or more of the authors are employed by commercial companies: FaunaPhotonics APS, Norsk Elektro Optikk AS.

3.1. Please provide an amended Funding Statement declaring this commercial affiliation, as well as a statement regarding the Role of Funders in your study. If the funding organization did not play a role in the study design, data collection and analysis, decision to publish, or preparation of the manuscript and only provided financial support in the form of authors' salaries and/or research materials, please review your statements relating to the author contributions, and ensure you have specifically and accurately indicated the role(s) that these authors had in your study. You can update author roles in the Author Contributions section of the online submission form.

3.2. Please also provide an updated Competing Interests Statement declaring this commercial affiliation along with any other relevant declarations relating to employment, consultancy, patents, products in development, or marketed products, etc. 

4. We note that S2 Figure in your submission contain satellite images which may be copyrighted. All PLOS content is published under the Creative Commons Attribution License (CC BY 4.0), which means that the manuscript, images, and Supporting Information files will be freely available online, and any third party is permitted to access, download, copy, distribute, and use these materials in any way, even commercially, with proper attribution. For these reasons, we cannot publish previously copyrighted maps or satellite images created using proprietary data, such as Google software (Google Maps, Street View, and Earth). For more information, see our copyright guidelines: http://journals.plos.org/plosone/s/licenses-and-copyright.

4.1.    You may seek permission from the original copyright holder of S2 Figure to publish the content specifically under the CC BY 4.0 license.

4.2.    If you are unable to obtain permission from the original copyright holder to publish these figures under the CC BY 4.0 license or if the copyright holder’s requirements are incompatible with the CC BY 4.0 license, please either i) remove the figure or ii) supply a replacement figure that complies with the CC BY 4.0 license. Please check copyright information on all replacement figures and update the figure caption with source information. If applicable, please specify in the figure caption text when a figure is similar but not identical to the original image and is therefore for illustrative purposes only.

Reviewers' comments:

Reviewer's Responses to Questions

**Comments to the Author**

1. Is the manuscript technically sound, and do the data support the conclusions?

Reviewer #1: Yes

Reviewer #2: Yes

2. Has the statistical analysis been performed appropriately and rigorously? 

Reviewer #1: I Don't Know

Reviewer #2: Yes

3. Have the authors made all data underlying the findings in their manuscript fully available?

Reviewer #1: Yes

Reviewer #2: Yes

4. Is the manuscript presented in an intelligible fashion and written in standard English?

Reviewer #1: Yes

Reviewer #2: Yes

5. Review Comments to the Author

Reviewer #1: This paper describes the use of lidar (laser radar) to estimate mosquito activity along a 589m transect extending from the edge of a rural Tanzanian village. The methods overall appear sound, the paper is well written and the science is creative and exhibits potential to better understand mosquito activity/behavior outside the houses and when mosquitoes are engaged in behaviors other than host seeking. Most of my comments are minor or are related to making this paper more understandable to those who are interested in mosquito behavior but lack knowledge/understanding of the system used to assess mosquito movement in this study (like me).

The only somewhat major comment is that the authors seem to largely skirt the question of what mosquitoes species they are detecting. Presumably, it is not possible to differentiate with 100% certainty using the lidar system but there was mention of trapping of mosquitoes at the time along with references to Culex (which seem to be the "high frequency females"), An. gambiae and An. funestus. However, details of the species distribution in this setting are not provided. Furthermore, is there any chance that these lower frequency detections were non-biting midges or Anopheles species that tend to feed on cattle and are therefore rarely detected in standard trapping methods? Could a larger Anopheles species (e.g. An. coustani) be confused with Culex?

1) Line 42. Suggest deleting the word "unprecedented" in this context. It is a bit of subtle bit of self-congratulatory praise that is inappropriate in scientific article.

2) Lines 46 & 47. Italicize Plasmodium and Anopheles.

3) Line 51-52. It should read "Unprecedented reductions...have averted" or "An unprecedented reduction...has averted".

4) Line 94 and Figure 1. The figure is a line graph, not a histogram.

5) Lines 98-99. The authors state that mosquitoes were detected closer on average to the village than other insects. However, other insects were those classified as larger than mosquitoes AND there is a decline in the detection due to distance. Is it possible that this difference is simply due to differences in the detection threshold for different sized insects? This is addressed again in lines 115-118. However, I did not see any attempt to estimate alpha for the different taxa in this study.

6) Lines 108-110. The discussion of fundamental wing-beat frequencies and modulation spectra was not apparent to this reader. I think this could use some more explanation in the text.

7) Line 115. "Nevertheless" is misspelled.

8) Lines 230-231. The authors speculate that males linger around the village as there are more nectar sources at the time of the measurements. While that is a possibility, is it also not a possibility that males remain close to the village as there are resting places, not to mention females in those resting places?

9) Lines 237-238. The authors state that activity near the village at dawn was likley bloodfed mosquitoes searching for oviposition sites. I assume they meant gravid females? Also, it seems a bit risky for females to venture out in the morning as light increases. One would think they are more likely to be predated upon and/or get stuck outside where high temperatures are likely to increase their chances of dying.

10) Line 244. The authors state that mosquitoes responsible for increased nightly activity observed soon after midnight may have been host-seeking An. funestus. I assume these mosquitoes were moving towards the village at that time? Also, how do they know it was An. funestus?

11) Line 271. I believe that WHO prefers the term "larval habitats" over "breeding sites".

12) Line 275. Similar to comment 10 above, why would the spike activity of parallel females after midnight be An. funestus?

13) Lines 285-286. The authors conclude that they "showed that male and female mosquitoes can be identified from their distinct wing-beat frequencies." This may have been shown in some of the references cited (e.g. reference 20) but this study did not generate data that would allow for the separation of males and females.

14) Line 312. The lasesr was focused onto a 2.5 x 23.3cm line on the termination target. Was that length by height? Based on the picture in Figure S2e, the height was 2.5 cm. Wouldn't it have made more sense to have rotated the beam 90 degrees? I would have thought you would capture more events in that orientation.

15) Line 328. The text indicates a probe volume of ~2m3 was monitored. Is this the 2.5cm x 23.3cm height and width of the beam multiplied by the lenght of the transect (589m)? Would be good to be explicit for the reader.

16) Line 346. For those not familiar with the lidar technology, it would be helpful to indicate what is meant by modulation spectra estimation so that the reader can understand more clearly why nearlyl half of all observations of insects transiting the beam were discarded.

17) Line 417. Similarly, it is unclear how these spectra were sorted by Euclidean distance. A bit more detail on how this was done would be helpful.

18) Line 421. Presumably, it should read "Male and female mosquitoes were differentiated from other insects..."

19) Line 444. The authors note that a female Anopheles weighs approximately 1.7mg. Is that An. gambiae or An. funestus? An. funestus are generally much smaller than An. arabiensis which is reported to be the primary Anopheles species detected in this site.

20) Line 629. The caption to figure S1 indicates that the beam was 102 mm in diameter. How does this correspond to the 2.5 cm x 23.3 cm measurements mentioned earlier in the manuscript?

21) Figure 3. I really like the creative display in Figure 3 but am wondering if there is a way to make the individual data points a bit more distinguishable from each other.

22) Figure 5. I noticed that the error bars do not correspond to what I would have expected. For example, the smallest confidence interval is for males during the day. Given that their numbers were very low during this period, it was unexpected that the confidence interval would be so tight. Any explanation for this?

Reviewer #2: In this manuscript the authors present an innovative tool and series of methods which may be standardized to monitor malaria vector populations using lidar.

This study is exciting as is the potential application of a lidar tool for malaria vector biology and control. Further, the in situ behavioral observations presented here have the opportunity to fill a large knowledge gap in terms of understanding high resolution spatiotemporal behavior of malaria vectors. Vector control tools are hypothesized to have altered feeding and resting behaviors and it is hypothesized that selective pressure on behaviors may exist. A standard behavioral assay has not been used with the implementation of vector control tools to understand whether these tools lead to behavioral resistance. With an "increase" in reported exophagy in many locations it seems as though there was a missed opportunity to determine the impact of vector control tools on behavioral selection. With the roll out of new vector control tools there is an opportunity to address this problem and not run into a similar issue with new vector control tools that target exophagy with the proposed lidar tool.

This manuscript will be a significant contribution in the literature and I look forward to reading more about this in the future

A few minor comments:

Introduction

LN 62: and baseline behavior of vector populations?

LN 69: please specify whether this is during a rainy or dry season

It is unclear if male mosquitoes refers to Anopheles spp. or mosquitoes in general. Were wingbeat frequencies determined in the field prior to this study? Is there geographic variation in wingbeat frequencies which may contribute to potential inaccuracies in insect determination?

Is there any way to combine this method with a collection strategy to further validate these data? There is mention of vehicle mounted sweep net drives, but is there another method which could be used throughout the lidar sampling period? sweep net drives may limit collection times.

Is there any way that this method can be used to improve collection strategies?

Similarly, can this approach be combined with new vector control tools like ATSBs to improve targeted approaches for exophagic mosquitoes?

The weather, weight, and feeding influences on wingbeat frequencies are significant, and the authors do describe the limitations of this, but it is concerning that these minor environmental factors could shift the interpretation of the data significantly. Further discussion and elaboration on these limitations in the discussion would be beneficial.

Is there an approach the authors can think of to classify local insect population wingbeats prior to the implementation of this method? Biological and environmental factors influencing wingbeats across taxa is my major concern with this method and other audio, wingbeat recording based vector tools, especially if frequencies are based on recordings from laboratory populations.

How could this tool be scalable across different nations where environmental factors will be variable?

Can this be used on a much smaller scale to develop behavioral assays classifying population-wide anthropophagic or zoophagic feeding preferences?

The reconstruction of host seeking behaviors is intriguing, but the interpretation seems as though a stretch with the assumption that high frequency females are hungry and host seeking. Is there variation in age structure and wingbeat?

The ecological contribution of potentially understanding temporal and spatial niche partitioning using this method across taxa are intriguing.

The crepuscular dispersal activity of mosquitoes here is really interesting, but to use these data for contradict HLC measurements may be a stretch given the potential margin of error in interpreting behaviors at the genus and species levels.

There is a lot of speculation in the discussion interpreting the behavioral observations, when it does not seem as though there is evidence for may of these (for example: Lns 237-24, Lns 264-268).

Is it possible to track individual mosquitoes using this technology?

6. PLOS authors have the option to publish the peer review history of their article (what does this mean?). If published, this will include your full peer review and any attached files.

Reviewer #1: No

Reviewer #2: No

---

## [Author Response · Author response to Decision Letter 0]

2 Dec 2020

To the Editor or PloS One, Dr. Daniel Becker

We thank you and the two reviewers for this opportunity to improve our manuscript and for the curiosity of the reviewers asking for details regarding our study. We acknowledge that the reviewers are highly knowledgeable on the topic and have correctly understood the purpose and implications of our manuscript. We appreciate their positive reception. Below we address the particular requests and points raised by the reviewers.

After some deliberation, and considering the journal is electronic, we would like to include the supplementary material in the manuscript. Would this be acceptable? The reviewers request a number of details which currently only appear in the supplementary file.

On behalf of all authors, sincerely 

Dr. Samuel Jansson 

Dept. Physics, Lund University

Reviewer #1

R1: This paper describes the use of lidar (laser radar) to estimate mosquito activity along a 589m transect extending from the edge of a rural Tanzanian village. The methods overall appear sound, the paper is well written and the science is creative and exhibits potential to better understand mosquito activity/behavior outside the houses and when mosquitoes are engaged in behaviors other than host seeking. Most of my comments are minor or are related to making this paper more understandable to those who are interested in mosquito behavior but lack knowledge/understanding of the system used to assess mosquito movement in this study (like me).

The only somewhat major comment is that the authors seem to largely skirt the question of what mosquitoes species they are detecting. Presumably, it is not possible to differentiate with 100% certainty using the lidar system but there was mention of trapping of mosquitoes at the time along with references to Culex (which seem to be the "high frequency females"), An. gambiae and An. funestus. However, details of the species distribution in this setting are not provided. Furthermore, is there any chance that these lower frequency detections were non-biting midges or Anopheles species that tend to feed on cattle and are therefore rarely detected in standard trapping methods? Could a larger Anopheles species (e.g. An. coustani) be confused with Culex? 

Authors: There have been a number of studies trying to estimate the classification accuracy. These studies are carried out in laboratories and only include a limited set of species and sexes. Such studies in very controlled conditions currently report accuracies in the range of 70-90%. The misclassified abundant ones could therefore obscure a rare species. Our trap catch included ~68% Anopheles s.l., and ~30% Culex s.p.p. The remaining 2% of mosquitoes caught, including An. funestus, An. coustani, Mansonia s.p.p. and Coquilettidia s.p.p., can be assumed to have very limited impact on the results. We did not observe any midges during the measurements, and large livestock is limited in the area where the study was conducted. Since midges are smaller than mosquitoes, they would also be less likely to be detected by the lidar system. In general, one could expect lower frequencies from larger insects, but many mosquito species select mates based on acoustics which leads to deviations from the general rule. 

Change: The trap catch data has been added to the manuscript (Table 1). The implications have been added to the discussion.

R1: Line 42. Suggest deleting the word "unprecedented" in this context. It is a bit of subtle bit of self-congratulatory praise that is inappropriate in scientific article. 

Authors: We appreciate this reminder to maintain a humble attitude. 

Change: The word “unprecedented” has been removed.

R1: Lines 46 & 47. Italicize Plasmodium and Anopheles. 

Change: This has been corrected.

R1: Line 51-52. It should read "Unprecedented reductions...have averted" or "An unprecedented reduction...has averted". 

Change: This has been corrected.

R1: Line 94 and Figure 1. The figure is a line graph, not a histogram. 

Authors: We somewhat disagree; the figure displays line graph presentations of histograms. Choosing lines graphs over bar plots allows us to display multiple lines, which would otherwise be obscured behind each other in bar plots. 

Change: The word “histograms” in line 94 has been replaced with “distributions”.

R1: Lines 98-99. The authors state that mosquitoes were detected closer on average to the village than other insects. However, other insects were those classified as larger than mosquitoes AND there is a decline in the detection due to distance. Is it possible that this difference is simply due to differences in the detection threshold for different sized insects? This is addressed again in lines 115-118. However, I did not see any attempt to estimate alpha for the different taxa in this study.

Authors: For the reason stated, it is challenging to report and interpret range profiles in entomological lidar in a quantitative and comparative manner. The present manuscript is a rare attempt to do just that. The different insect taxa are represented by the “Others” group in the manuscript, and alpha is presented for this group. Presenting individual alpha values for more groups in a digestible manner for the reader is challenging and not within the scope of this manuscript. The difference in alpha values cannot solely be explained by the range bias outlined by R1 for a few reasons. Not all the low frequent clusters are larger than mosquitoes (See Fig. S9), and for example in Fig.5a-Day 8:40-17:00 the other low frequent insects are significantly (95% conf.) closer than high-frequent females. As such, during this time interval, mosquitoes are detector further away than the low frequent flyers. 

Change: We have emphasized that the difference in alpha values between mosquitoes and other insects could not solely be explained by detection range limits. The caption of Figure 1 now states that mosquitoes being observed closer to the village than other insects may in part be attributed to the decreasing sensitivity with range.

R1: Lines 108-110. The discussion of fundamental wing-beat frequencies and modulation spectra was not apparent to this reader. I think this could use some more explanation in the text.

Authors: We apologize for this inconvenience. These terms are explained in the Methods section, which was originally included as supplementary material further down in the manuscript. We have now moved the methods section into the main manuscript, before the Results section, and the reader should therefore be acquainted with the terms upon reaching Figure 2. 

Change: The Methods section has been moved into the main manuscript, and the mentioned sentence has been reworded.

R1: Line 115. "Nevertheless" is misspelled. 

Change: This has been corrected. 

R1: Lines 230-231. The authors speculate that males linger around the village as there are more nectar sources at the time of the measurements. While that is a possibility, is it also not a possibility that males remain close to the village as there are resting places, not to mention females in those resting places? 

Authors: Agreed. 

Change: Interpretations added. 

R1: Lines 237-238. The authors state that activity near the village at dawn was likley bloodfed mosquitoes searching for oviposition sites. I assume they meant gravid females? Also, it seems a bit risky for females to venture out in the morning as light increases. One would think they are more likely to be predated upon and/or get stuck outside where high temperatures are likely to increase their chances of dying. 

Authors: We have previously shown by lidar recordings at a tropical site in China1, that predation is minimal at certain temporal niches during dusk and dawn, between the visual predation of swifts and the acoustic predation by bats. Surprisingly, we found no evidence of vertebrate predators in the present dataset. We agree with R1 that water evaporation due to high temperature is a significant threat. However, at 6:00 morning, the temperature is on the minimum of 21 °C and relative humidity is at its maximum of 70% (for details see also ref. 44). Therefore, dawn poses minimal risk in terms of both predation and evaporation.

R1: Line 244. The authors state that mosquitoes responsible for increased nightly activity observed soon after midnight may have been host-seeking An. funestus. I assume these mosquitoes were moving towards the village at that time? Also, how do they know it was An. funestus? 

Authors: At the moment, we are not able to differentiate An. funestus from An. arabiensis in the present data set. However, An. funestus is known to have a different circadian rhythm (they host-seek after midnight2) compared to the An. gambiae subspecies including An. arabiensis, which is the more abundant species in Lupiro. 

Change: We have clarified that An. funestus and An. arabiensis could not be differentiated at the moment in the present lidar dataset. 

R1: Line 271. I believe that WHO prefers the term "larval habitats" over "breeding sites".

Authors: We thank R1 for pointing this out. 

Change: The term "breeding sites" was changed to "larval habitats".

R1: Line 275. Similar to comment 10 above, why would the spike activity of parallel females after midnight be An. funestus? 

Authors: As mentioned above, this is our speculation based on the known biting behavior of An. funestus in the area.

R1: Lines 285-286. The authors conclude that they "showed that male and female mosquitoes can be identified from their distinct wing-beat frequencies." This may have been shown in some of the references cited (e.g. reference 20) but this study did not generate data that would allow for the separation of males and females. 

Authors: We agree with R1 that this is not what we showed in the present study. Rather, we demonstrated that male and female mosquitoes can be distinguished in field conditions with lidar, based on laboratory reference measurements of wing-beat frequency. 

Change: The mentioned lines have been reworded and now accurately describe what was shown.

R1: Line 312. The lasesr was focused onto a 2.5 x 23.3cm line on the termination target. Was that length by height? Based on the picture in Figure S2e, the height was 2.5 cm. Wouldn't it have made more sense to have rotated the beam 90 degrees? I would have thought you would capture more events in that orientation. 

Authors: Our apologies for being unclear, the stated values are height by width. Scheimpflug lidar requires that polarization is parallel with the lidar baseline (the tilted detector works best for this polarization). Also the elongated image of the diode laser source (Fig.S2e) needs to be perpendicular to the lidar baseline (vertical aluminum bar in Fig.S2g) to match the elongated pixel footprints. It is therefore a matter choosing either vertical or horizontal baseline. The system was constructed a few months before the campaign and the experience of such systems were limited at the time. The argument for choosing a vertical baseline was that insect wings would transmit at the Brewster angle and that this could produce more detailed waveforms. This is mainly a theoretical speculation however.

We do not agree that more events would be captured by orienting the lidar baseline horizontally. We can assume that most insect transport is horizontal but the probe volume of the system is the spatial product of both laser illumination and detection field. With the given laser, beam expander, sensor and receiver telescope, the probe volume is 12 cm tall and 0.75cm wide at 30 m distance, and 2.5 cm tall and 18 wide @at 598 m distance. Since the sensitivity is highest at close range, we hypothesize that a horizontal baseline would produce:

• Lower overall counts

• Fewer counts at close range

• More counts at long range

• Longer observations at close range

• Shorter observations at long range

• Overall better frequency resolution 

We do agree that it would be interesting to investigate how the orientation of the baseline affects the range distribution and alpha value. At the moment we do not have data for such analysis. 

Change: We have clarified the height and width of the probe volume at both close and far range. We have noted that orientation of the lidar baseline can affect number of observations and their transit time (and thus frequency resolution). 

R1: Line 328. The text indicates a probe volume of ~2m3 was monitored. Is this the 2.5cm x 23.3cm height and width of the beam multiplied by the lenght of the transect (589m)? Would be good to be explicit for the reader. 

Authors: The illuminating beam shaped approximately like a toothpaste tube, starting out circular with a diameter ø12.7 cm at the transmission telescope and ending as a 2.5x23.3 cm oval at the termination target. The probe volume is also a product of the pixel footprints. The dimension of the probe volume is stated in the previous reply. We have derived the value numerically by estimating the volume of each probe volume voxel and integrated the region of interest. A coarse estimate can also be derived by multiplying the mean height and width and the length of the region of interest, as suggested by R1 here. Note that the system had a near limit of 35 m and that some margin was required between insect echoes and the termination. Therefore, the probe volume is shorter than the range to the termination. 

Change: We have clarified the dimensions of the probe volume, also in relation to the previous comment.

R1: Line 346. For those not familiar with the lidar technology, it would be helpful to indicate what is meant by modulation spectra estimation so that the reader can understand more clearly why nearlyl half of all observations of insects transiting the beam were discarded. 

Authors: These aspects relate more to the mathematics of Fourier transforms and practicalities of DSP (digital signal processing) than to lidar. The observable frequency range extends from the inverse of the time duration of the insect signal to the Nyquist frequency (half of the sample rate, 875 Hz in our system after background subtraction). This is because enough time to observe at least one period is needed to verify a low frequency, and because at least two samples per period are needed to verify a high frequency. The frequency response of our instrument is the normalized sinc(f/fNq). This implies high frequencies and insect harmonics are subject to signal processing artifacts such as attenuation, phase delay, beating and folding. The minimum time duration of insect signals also determines the frequency resolution, which is the reason for excluding short signals. In conclusion, we could have chosen to include more insect observations, yielding a shorter observable frequency range and lower resolution, or we could have chosen to include less observations and gained higher frequency resolution and observability for lower frequencies. 

Change: We have expanded the section and elaborated on these aspects and details.

R1: Line 417. Similarly, it is unclear how these spectra were sorted by Euclidean distance. A bit more detail on how this was done would be helpful. 

Authors: After estimating the modulation power, each insect observation is represented by 40 parameters, corresponding to the modulation power at the observed frequencies. Our expectation is that similar insects will have similar wing beats and therefore similar modulation spectra. Hierarchical clustering implies calculating the statistical similarity between all possible pairs of observations in this 40-dimensional parameter space. Euclidean distance is one metric commonly used to calculate this statistical similarity, and is the multi-dimensional expansion of the Pythagorean theorem. 

Change: We have expanded the details regarding hierarchical clustering. 

R1: Line 421. Presumably, it should read "Male and female mosquitoes were differentiated from other insects..." 

Authors: Agreed. 

Change: Corrected. 

R1: Line 444. The authors note that a female Anopheles weighs approximately 1.7mg. Is that An. gambiae or An. funestus? An. funestus are generally much smaller than An. arabiensis which is reported to be the primary Anopheles species detected in this site. 

Authors: This is an An. arabiensis. 

Change: This is now clearly written in the mentioned sentence. 

R1: Line 629. The caption to figure S1 indicates that the beam was 102 mm in diameter. How does this correspond to the 2.5 cm x 23.3 cm measurements mentioned earlier in the manuscript? 

Authors: Our apologies, but the expander is in fact ø127 mm and not ø102 mm. Explanation regarding beam shape was provided according to earlier request. 

Change: We have corrected value for beam expander. As per earlier comment, we have provided details about beam shape and probe volume.

R1: Figure 3. I really like the creative display in Figure 3 but am wondering if there is a way to make the individual data points a bit more distinguishable from each other. 

Authors: We thank R1 for his compliments. We are afraid that increasing the point size would smear observation together and not facilitate differentiation. Also diminishing point size seem to reduce visibility. 

R1: Figure 5. I noticed that the error bars do not correspond to what I would have expected. For example, the smallest confidence interval is for males during the day. Given that their numbers were very low during this period, it was unexpected that the confidence interval would be so tight. Any explanation for this? 

Authors: It is true that a range distribution with fewer observations is prone to subject to more randomness. We note that the day interval is longer than the other intervals. The confidence interval for alpha reflects how well the recorded distribution can be described by the simple power law. For example, for the time interval including male swarm (Fig.3) which could not be explained by the simple power law, the confidence interval is consequently larger. 

Change: We have added a remark emphasizing the meaning of the confidence interval for alpha.

Reviewer #2

In this manuscript the authors present an innovative tool and series of methods which may be standardized to monitor malaria vector populations using lidar. 

This study is exciting as is the potential application of a lidar tool for malaria vector biology and control. Further, the in situ behavioral observations presented here have the opportunity to fill a large knowledge gap in terms of understanding high resolution spatiotemporal behavior of malaria vectors. Vector control tools are hypothesized to have altered feeding and resting behaviors and it is hypothesized that selective pressure on behaviors may exist. A standard behavioral assay has not been used with the implementation of vector control tools to understand whether these tools lead to behavioral resistance. With an "increase" in reported exophagy in many locations it seems as though there was a missed opportunity to determine the impact of vector control tools on behavioral selection. With the roll out of new vector control tools there is an opportunity to address this problem and not run into a similar issue with new vector control tools that target exophagy with the proposed lidar tool. 

This manuscript will be a significant contribution in the literature and I look forward to reading more about this in the future. 

R2: LN 62: and baseline behavior of vector populations? 

Change: Added.

R2: LN 69: please specify whether this is during a rainy or dry season 

Authors: This is during the cool dry season, there is no precipitation and virtually no wind during recordings.

Change: Information added.

R2: It is unclear if male mosquitoes refers to Anopheles spp. or mosquitoes in general. Were wingbeat frequencies determined in the field prior to this study? Is there geographic variation in wingbeat frequencies which may contribute to potential inaccuracies in insect determination? 

Authors: The reported mosquitoes are assumed to be 74% Anopheline and 24% Culicine according to catch analysis. Yes, there are several factors which affect wingbeat frequency: the ambient temperature, gravidity stage and wing length. Wing length3 can differentiate between locations and there are systematic differences between laboratory cultures and wild populations. Based on literature and our own laboratory experiments we know that there is significant overlap in wing-beat frequency between Anopheline and Culicine mosquitoes of the same sex, but not between sexes. It is important to understand that Hierarchical Cluster Analysis (HCA) is a top-down method that groups observations based on statistical similarity. With the number of clusters selected in this study, it was not possible to distinguish males of different species, whereas we could make educated guesses on the female species. 

Change: We have clarified that the group of male mosquitoes likely corresponds to a mixture of Anopheline and Culicine mosquitoes.

R2: Is there any way to combine this method with a collection strategy to further validate these data? There is mention of vehicle mounted sweep net drives, but is there another method which could be used throughout the lidar sampling period? sweep net drives may limit collection times.

Authors: This is speculative, but we believe truck based sweep nets are a good and representative candidate for ground truthing the lidar probe volume. There are at least a dozen different mosquito trapping methods, including rotational auto traps for sectioning the hours, but each trap type is limited to particular mosquito species, sexes, night hours or life stages such as swarming, gravid, host seeking, egg laying etc… Lidar could be combined with any field-sampling method, but we mention sweep net drives specifically because we believe it will produce an unbiased result which is crucial for interpreting the lidar data. 

R2: Is there any way that this method can be used to improve collection strategies? 

Authors: Absolutely, as R2 would be aware, essentially any type of trap for mosquito surveillance, is highly sensitive to the exact positioning in landscape such as the vicinity of water bodies, host plants or animals, shade at different hours as well as topography. As demonstrated in e.g. Fig.1 & 3, the activity can be confined to particular land marks. It is possible to sweep the entomological lidar beam slowly over a field and produce a radial map of activity4, this could provide an landscape overview, e.g., for the purpose of strategically positioning surveillance traps. 

Change: This is now mentioned in the manuscript. 

R2: Similarly, can this approach be combined with new vector control tools like ATSBs to improve targeted approaches for exophagic mosquitoes? 

Authors: As indicated previously, lidar systems may be used to scan larger areas and provide landscape-level surveillance of vectors. They could then be used to improve the geopositioning of supplementary malaria vector control interventions such as attractive targeted sugar baits (ATSBs) or odor-based traps. Previously, it has been shown that targeting such interventions by following the Pareto principle, i.e. 80/20 statistical rule, could vastly improve the epidemiological outcomes while reducing the amount of effort and resources required. Lidar-based surveillance platforms could be used to rapidly scan entire villages and estimate areas with high Anopheles densities, thereby improving such geo-targeting of supplementary interventions. 

Change: This is now mentioned in the manuscript. 

R2: The weather, weight, and feeding influences on wingbeat frequencies are significant, and the authors do describe the limitations of this, but it is concerning that these minor environmental factors could shift the interpretation of the data significantly. Further discussion and elaboration on these limitations in the discussion would be beneficial. 

Authors: It is true that the influences of temperature, weight and e.g. laboratory cultures vs. wild populations can be measured and quantified to significantly different effect from zero. However, such WBF shift can account for Δ50Hz as opposed to difference between sexes of 165-330 Hz5. The weather during our recordings was stable and reproducible from day to day. For instance, we estimate a frequency shift of Δ50Hz for Anopheline mosquitoes and Δ30 Hz for Culicine mosquitoes from morning to evening rush hours. More importantly, we are not imposing any hard frequency intervals in the data using prior knowledge. We used the entirely objective HCA to detect a number of differentiable clusters; we then identified the fundamental tone of the centroid modulation spectra a derived the plausible origin among the present bulk species. 

Change: We have moved the discussion on the impact of weather and weight from the supplementary material to the discussion section of them manuscript.

R2: Is there an approach the authors can think of to classify local insect population wingbeats prior to the implementation of this method? Biological and environmental factors influencing wingbeats across taxa is my major concern with this method and other audio, wingbeat recording based vector tools, especially if frequencies are based on recordings from laboratory populations. 

Authors: We agree that accurate characterization of classified insect species can facilitate interpretation of lidar data. We have complemented these field measurements with various laboratory reference measurements. These include ex vivo estimation of scattering cross sections and depolarization ratio by imaging goniometry on both dry and fresh tropical mosquito species6. We are currently working on reducing the size of this instrument to bring in to tropical field sites. Ex vivo measurements suffer from the inability to provide the dynamic wing-beat properties exploited here. Both our group7 and others8,9 have built rather simple setups to retrieve in vivo wing beat modulation spectra from known species in enclosed chambers. In principle such devices could be brought into the field for in situ recordings (this was done by FaunaPhotonics in Rothamstad, submitted work). Much of these efforts are reported in the doctoral thesis of the first author10. We have attempted to do controlled releases of known species in the lidar beam. However, these experiments require access to the beam which we normally keep inaccessible over ground for eye-safety reasons. These attempts are fairly complicated and with a very low fraction of successful recordings because the beam is narrow and released individuals could fly in any direction. Finally, there are indirect correlations with trap catches, but as we know these are subject to biases of all kinds. Our experience with the acoustic Humbug project from Oxford is limited. We have briefly attempted to record WBFs from captured insects in small netted cups using cellphone audio recordings, trapped bees tend to produce very different frequencies when trapped. 

We would like to point out that the current study not only consider WBFs but the whole modulation spectrum, which includes information such as body-to-wing ratios and harmonic overtones relating e.g. to the glossiness of the wings. 

Change: Could we say that we added some discussion about ex-vivo, in-vivo and in-situy referencing, controlled releases and trap correlations? Just some blabla? Are you citing you own thesis? – would be highly appropriate with all the nice details in the intro.

R2: How could this tool be scalable across different nations where environmental factors will be variable?

Authors: Scaling up the usage would require either commercializing and production of the instruments in large numbers, or alternatively sharing design, part list and processing code on open access, and we are supporting both approaches within our capabilities. The clustering we have applied is unsupervised and not based on any a priori knowledge. No doubt, data from different locations and/or seasons would produce a different number of clusters with distinct centroid modulation spectra and wing-beat frequencies. While existing literature on mosquito wingbeat frequencies and their thermal shift can allow qualified guesses on coarse taxonomic groups, better interpretation would require knowledge of the sites and verification by trapping and identification.

R2: Can this be used on a much smaller scale to develop behavioral assays classifying population-wide anthropophagic or zoophagic feeding preferences? 

Authors: Yes, Scheimpflug lidar can be rescaled to alter range and resolution11. We have experience in laboratory applications in combustion diagnostics and aquaculture with a measurement range of a few meters, in drone-mounted lidar measurements of forest canopy health covering a few tens of meters, and in landscape-scale measurements over several kilometers. The requirements of the application have implications on the design of the lidar system, so the same instrument would not be appropriate for all detection distances.

R2: The reconstruction of host seeking behaviors is intriguing, but the interpretation seems as though a stretch with the assumption that high frequency females are hungry and host seeking. Is there variation in age structure and wingbeat? 

Authors: We are not aware of an age-dependency of wing-beat frequency. Other optical properties are known to vary with age12, and we speculate that this could relate to refractive index. At the moment, optical assessment of mosquito ages is not fully explored. We agree with R2 that the reconstruction of behavior should not be stretched too far – it is, after all, our interpretation of the signals based on known behaviors. This is indicated in all places where it is mentioned.

R2: The ecological contribution of potentially understanding temporal and spatial niche partitioning using this method across taxa are intriguing. 

Authors: We appreciate this acknowledgement by R2.

R2: The crepuscular dispersal activity of mosquitoes here is really interesting, but to use these data for contradict HLC measurements may be a stretch given the potential margin of error in interpreting behaviors at the genus and species levels. 

Authors: We agree with R2 that the lidar-based observations are unlikely to be directly representative of HLC catches since these would be caught in different physiological states. The HLC catches are likely host-seeking, whereas lidar-observed mosquitoes are likely homing, mating or swarming. Rather than contradicting HLC measurements, lidar-based observations are useful for overall assessment of population densities and their concentrations in differenct locations. 

Change: This is now elaborated in the discussion.

R2: There is a lot of speculation in the discussion interpreting the behavioral observations, when it does not seem as though there is evidence for many of these (for example: Lns 237-24, Lns 264-268). 

Change: We have softened the wording in several places in the manuscript to indicate clearly when our interpretation of the data is speculative.

R2: Is it possible to track individual mosquitoes using this technology? 

Authors: In its current state, entomological lidar is not capable of tracking individual insects. This could in theory be possible, but would require vast improvements in engineering and real-time data processing. In addition, the signal quality might decrease due to mechanical vibration of the system, and eye-safety would be a large concern with a moving beam. Other groups13-15 have demonstrated tracking of individual mosquitoes indoors on short scale with other methods.

References

1 Malmqvist, E. et al. The bat-bird-bug battle: daily flight activity of insects and their predators over a rice field revealed by high resolution Scheimpflug Lidar Royal Society Open Science 5 (2018).

2 Limwagu, A. J. et al. Using a miniaturized double-net trap (DN-Mini) to assess relationships between indoor-outdoor biting preferences and physiological ages of two malaria vectors, Anopheles arabiensis and Anopheles funestus. Malar J 18, 282, doi:10.1186/s12936-019-2913-9 (2019).

3 Faiman, R. et al. Quantifying flight aptitude variation in wild Anopheles gambiae in order to identify long-distance migrants. Malaria Journal 19, 1-15 (2020).

4 Tauc, M. J., Fristrup, K. M., Repasky, K. S. & Shaw, J. A. Field demonstration of a wing-beat modulation lidar for the 3D mapping of flying insects. OSA Continuum 2, 332-348, doi:10.1364/OSAC.2.000332 (2019).

5 Jansson, S., Gebru, A., Ignell, R., Abbott, J. & Brydegaard, M. in SPIE/OSA European Conferences on Biomedical Optics.

6 Jansson, S., Atkinson, P., Ignell, R. & Brydegaard, M. First Polarimetric Investigation of Malaria Mosquitos as Lidar Targets. IEEE JSTQE Biophotonics 25, 1-8 (2018).

7 Gebru, A. et al. Multiband modulation spectroscopy for determination of sex and species of mosquitoes in flight. J. Biophotonics 11 (2018).

8 Genoud, A. P., Gao, Y., Williams, G. M. & Thomas, B. P. Identification of gravid mosquitoes from changes in spectral and polarimetric backscatter cross‐sections. Journal of biophotonics, e201900123 (2019).

9 Potamitis, I., Rigakis, I., Vidakis, N., Petousis, M. & Weber, M. Affordable bi-modal optical sensors to spread the use of automated insect monitoring. Journal of Sensors, 25 (2018).

10 Jansson, S. Entomological Lidar: Target Characterization and Field Applications, Lund University, (2020).

11 Malmqvist, E., Brydegaard, M., Aaldén, M. & Bood, J. Scheimpflug Lidar for combustion diagnostics. Opt. Express (2018).

12 Lambert, B. et al. Monitoring the age of mosquito populations using near-infrared spectroscopy. Scientific reports 8, 5274 (2018).

13 Butail, S. et al. Reconstructing the flight kinematics of swarming and mating in wild mosquitoes. Journal of The Royal Society Interface 9, 2624-2638, doi:10.1098/rsif.2012.0150 (2012).

14 Parker, J. E. et al. Infrared video tracking of Anopheles gambiae at insecticide-treated bed nets reveals rapid decisive impact after brief localised net contact. Scientific reports 5, 13392 (2015).

15 Mullen, E. R. et al. Laser system for identification, tracking, and control of flying insects. Opt. Express 24, 11828-11838, doi:10.1364/OE.24.011828 (2016).

---

## [Decision Letter · Decision Letter 1]

15 Feb 2021

Real-time dispersal of malaria vectors in rural Africa monitored with lidar

PONE-D-20-15981R1

Dear Dr. Jansson,

We’re pleased to inform you that your manuscript has been judged scientifically suitable for publication and will be formally accepted for publication once it meets all outstanding technical requirements.

Kind regards,

Daniel Becker

Academic Editor

PLOS ONE

Additional Editor Comments (optional):

I thank the authors for their patience with the review process and for their thorough revision on this interesting contribution. Please address the one minor comment from the reviewer in your final manuscript submission.

Reviewers' comments:

Reviewer's Responses to Questions

**Comments to the Author**

1. If the authors have adequately addressed your comments raised in a previous round of review and you feel that this manuscript is now acceptable for publication, you may indicate that here to bypass the “Comments to the Author” section, enter your conflict of interest statement in the “Confidential to Editor” section, and submit your "Accept" recommendation.

Reviewer #2: All comments have been addressed

2. Is the manuscript technically sound, and do the data support the conclusions?

Reviewer #2: Yes

3. Has the statistical analysis been performed appropriately and rigorously? 

Reviewer #2: Yes

4. Have the authors made all data underlying the findings in their manuscript fully available?

Reviewer #2: Yes

5. Is the manuscript presented in an intelligible fashion and written in standard English?

Reviewer #2: Yes

6. Review Comments to the Author

Reviewer #2: This is an interesting contribution to the literature. This is a fascinating approach and I look forward to the further development of this new direction of research.

In the response to reviewers document there is one place where authors may have left in some text among the authors (see below). Other than that all comments were addressed completely and I appreciate the thoroughness of the responses and inclusion of supportive references.

"Change: Could we say that we added some discussion about ex-vivo, in-vivo and insituy referencing, controlled releases and trap correlations? Just some blabla? Are you citing you own thesis? – would be highly appropriate with all the nice details in the intro."

7. PLOS authors have the option to publish the peer review history of their article (what does this mean?). If published, this will include your full peer review and any attached files.

Reviewer #2: No

---

## [Editor Report · Acceptance letter]

23 Feb 2021

PONE-D-20-15981R1 

Real-time dispersal of malaria vectors in rural Africa monitored with lidar 

Dear Dr. Jansson:

I'm pleased to inform you that your manuscript has been deemed suitable for publication in PLOS ONE. Congratulations! Your manuscript is now with our production department. 

Kind regards, 

on behalf of

Dr. Daniel Becker 

Academic Editor

PLOS ONE